# The Shape of Adversarial Influence: Characterizing LLM Latent Spaces with Persistent Homology

**Aideen Fay**[*][†]
Microsoft Security Response Center

**Inés García-Redondo**[*][†]
AIDOS Lab, University of Fribourg

**Qiquan Wang**[*][†]  **Haim Dubossarsky**[‡]
Queen Mary University of London

**Anthea Monod**[†]
Imperial College London

## Abstract

Existing interpretability methods for Large Language Models (LLMs) predominantly capture linear directions or isolated features. This overlooks the high-dimensional, relational, and nonlinear geometry of model representations. We apply persistent homology (PH) to characterize how adversarial inputs reshape the geometry and topology of internal representation spaces of LLMs. This phenomenon, especially when considered across operationally different attack modes, remains poorly understood. We analyze six models (3.8B to 70B parameters) under two distinct attacks, indirect prompt injection and backdoor fine–tuning, and show that a consistent topological signature persists throughout. Adversarial inputs induce topological compression, where the latent space becomes structurally simpler, collapsing the latent space from varied, compact, small-scale features into fewer, dominant, large-scale ones. This signature is architecture-agnostic, emerges early in the network, and is highly discriminative across layers. By quantifying the shape of activation point clouds and neuron-level information flow, our framework reveals geometric invariants of representational change that complement and extend existing linear interpretability methods.

## 1 Introduction

Understanding how adversarial conditions alter the internal representations of Large Language Models (LLMs) requires tools that can characterize complex, high-dimensional structure. Traditional interpretability approaches, such as linear probes or activation-based feature extraction, focus on linearly extractable or isolated directions in latent space (Alain & Bengio, 2016; Zou et al., 2023; Cunningham et al., 2023), which can overlook relational, nonlinear, and global geometric structure (Engels et al., 2025; Gebhart et al., 2019; Li et al., 2024b). This limitation has practical implications for the robustness of safeguards based on these methods, especially in scenarios where representation geometry meaningfully influences model behavior under perturbation (Ilyas et al., 2019; Kirch et al., 2024; Wehner et al., 2025). Moreover, existing studies often examine a single class of adversarial behavior in isolation, leaving open the more fundamental question of whether attacks arising through qualitatively different mechanisms imprint a shared signature on the model's internal state.

In this paper, we report progress on both fronts. First, we argue that *persistent homology* (PH), the workhorse of topological data analysis (TDA), is uniquely suited to interpreting LLM representations. PH is provably robust to noise (Cohen-Steiner et al., 2007) and provides a coordinate-free, multiscale summary of the relational geometry within a latent space. Unlike methods that project high-dimensional representations onto lower-dimensional subspaces, PH captures multi-scale topological structure through a filtration that encodes both local clustering patterns and global topological

---

[*]Equal contribution. Corresponding author: `aideen.fay23@imperial.ac.uk`
[†]Department of Mathematics, Imperial College London.
[‡]Also affiliated with: Language Technology Lab, University of Cambridge; The Alan Turing Institute.

features, enabling direct comparison across models, input distributions, and fine-tuning stages. This information is encoded in a *barcode*, a compact summary of how topological features evolve across the filtration. As shown in Figure 1, these barcodes immediately reveal a clear distinction between the activations produced by normal and adversarial inputs. Second, we apply PH to two fundamentally distinct attack vectors. Indirect prompt injection (Greshake et al., 2023) exploits the model's inability to separate instructions from data at inference time, while sandbagging (van der Weij et al., 2024) via backdoor fine-tuning (Hubinger et al., 2024) manipulates the training process itself to induce intentional underperformance when triggered by a specific input.

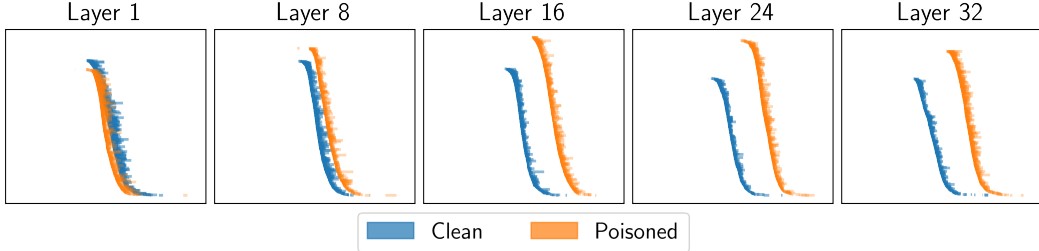

Figure 1: **Example barcodes from clean vs. poisoned activations.** PH of two samples of $n = 1000$ activations of clean (blue) and poisoned (orange) activations of Mistral 7B at each of the 5 layers (1, 8, 16, 24, 32).

Our contributions can be summarized as follows.

- We comprehensively analyze six state-of-the-art models under two fundamentally different attack modes. We show that adversarial inputs induce *consistent topological behavior within the LLM latent space*. Specifically, adversarial inputs cause latent representations to become more dispersed, characterized by fewer but more topologically significant large-scale features. In contrast, normal inputs produce a greater diversity of compact, small-scale structures.

- We show that this phenomenon *holds across models ranging from 7B to 70B parameters*, suggesting that adversarial triggers systematically reshape the representation space in a consistent and predictable manner that is independent of specific architectures or training procedures.

- We introduce a novel, *neuron-level PH analysis* that confirms these geometric shifts at a finer scale, revealing a *phase transition in the topological complexity* of the information flow.

While standard linear probes can also separate normal from adversarial activations with high accuracy, they do not explain *what* distinguishes these representations geometrically. Our topological framework fills this gap by revealing the structural basis of this separability and, because PH summaries are directly comparable, enabling uniform characterization of attacks that differ in origin but may share a latent geometric signature.

## 2 BACKGROUND

In this section, we introduce the reader to PH and motivate the two adversarial scenarios we focus on throughout this work.

### 2.1 PERSISTENT HOMOLOGY AND PERSISTENCE BARCODES

PH is a technique used to quantify the "shape" and "size" of data. It captures higher-order relational information and has an inherently interpretable nature. More precisely, PH captures *topological features*, e.g., connected components, tunnels and loops, or cavities and bubbles, present at different scales in our data.

For our activation data, i.e., point clouds $X \subset \mathbb{R}^D$, with $D$ the hidden dimension of the model (typically, $D = 4096$) and where each point is the latent representation of the last token in a prompt in a given layer; the PH pipeline proceeds as follows. The first step is to construct a dynamic, geometric representation of our point cloud. A classical construction involves the *Vietoris–Rips* complex, which for a scale parameter $\epsilon > 0$ is obtained from the $\epsilon$-*neighborhood graph*, that is, the graph where we connect any two points at distance less than $\epsilon$. The Vietoris–Rips complex goes beyond the pairwise interactions in the $\epsilon$-neighborhood graph including higher-order relational information, namely, interactions between more than two points at the same time, known as *simplices*: 0-simplices correspond to points, 1-simplices to edges, 2-simplices to triangles, 3-simplices to tetrahedra, and so on. We add a simplex between a subset of 3 or more points to the Vietoris–Rips complex whenever they are all pairwise connected, for instance, we add a triangle if three points in the point cloud are connected in the $\epsilon$-neighborhood graph. This completes the Vietoris–Rips complex construction. Considering all scale parameters $\epsilon$ at the same time, we obtain the *Vietoris–Rips filtration*: a growing family of geometric spaces where we connect points and add simplices as the parameter $\epsilon$ grows.

PH then leverages algebraic topology to produce the *persistence barcode*, a collection of bars capturing how the topological features are formed and disappear in the filtration as the scale parameter $\epsilon$ increases. The barcode is stratified in different dimensions, here we focus in dimensions 0 and 1. Bars in the 0-dimensional barcode (or 0-bars) correspond to connected components: at $\epsilon = 0$ there are as many bars in the barcode as points in the data, with bars terminating as point get connected in the $\epsilon$-graph. 1-bars represent loops or cycles in the corresponding Vietoris–Rips complex: a bar starts whenever we have added enough edges to enclose a non-trivial hole, and ends when the addition of triangles covers said hole. Usually, the starting point is called the *birth* and the ending point the *death* of the bar. An illustrative example of the PH pipeline in a simple point cloud and the corresponding barcode can be found in Figure 2. See Appendix A.1 for more details on the PH construction.

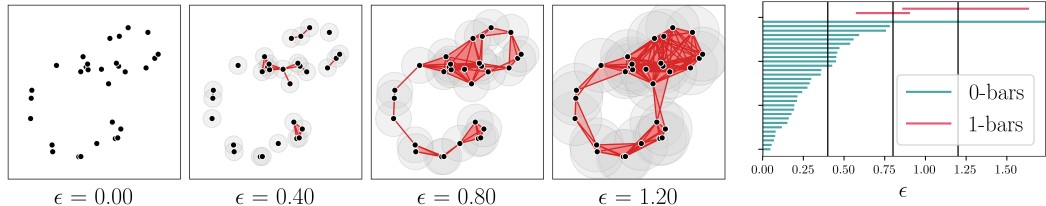

$\epsilon = 0.00 \qquad \epsilon = 0.40 \qquad \epsilon = 0.80 \qquad \epsilon = 1.20$

Figure 2: **Left:** Vietoris–Rips filtration constructed from a sample of 50 points over 2 circles with noise, at four values of the distance threshold $\epsilon \in [0, \infty)$. **Right:** corresponding persistence barcode for the 0- and 1-bars, with vertical lines corresponding to the thresholds displayed on the left.

## 2.2 ADVERSARIAL INFLUENCE ON LLMS

PH has been applied to earlier deep learning architectures (e.g., CNNs and MLPs) (Naitzat et al., 2020; Zhang et al., 2024), and a recent survey by Sekuloski et al. (2026) charts the evolving research landscape of TDA for understanding LLM behavior. For example, Gardinazzi et al. (2024) use zigzag persistence to track the evolution of topological features across transformer layers and define a persistence similarity measure that supports topology-aware layer pruning.

To the best of our knowledge, our work is the first to scale and systematically use PH to characterize large-scale LLM activation spaces (across models up to 80B parameters) under a range of adversarial interventions. Furthermore, we show that the resulting topological signatures yield complementary interpretability benefits for understanding adversarial influence on representations.

In order to test the generality of our approach, we quantify and interpret the effects of two systematically different attack modes which carry high security impact—*Indirect Prompt Injection (XPIA)*, where attackers embed hidden instructions in retrieved content to override a user's original prompt (Greshake et al., 2023; Rehberger, 2024); and *sandbagging via backdoor fine-tuning*, which involves deliberately training a model to suppress its capabilities until a secret trigger is provided (Greenblatt et al., 2024; van der Weij et al., 2024). These techniques target fundamentally

distinct vulnerabilities: XPIA exploits the model's core inability to distinguish data from instructions (Zverev et al., 2025), whereas sandbagging affects the fine-tuning process.

## 2.3 PERSISTENT HOMOLOGY IN MACHINE LEARNING: BARCODE SUMMARIES

Persistence barcodes cannot be directly used as input features in a ML model since they do not reside in a Euclidean space (Turner et al., 2014). We circumvent this issue by studying summary statistics of barcodes (Ali et al., 2023)—such as the mean, standard deviation, median, or quartiles—of the empirical distributions of the births, deaths, and *persistences* (lengths) of the bars in a given barcode. We can also study the empirical distribution of the ratios between births and deaths, which have the advantage of being scale invariant; the number of bars, providing a notion of topological diversity; the total persistence, which is given by the sum of the lengths of all bars in the barcode and captures both the number of topological features and their size; and the *persistent entropy* (Chintakunta et al., 2015; Rucco et al., 2016) of each barcode, which intuitively measures the heterogeneity within the lengths of the bars in the barcode. In all, for each barcode, we compute a 41-dimensional descriptive feature vector that can be used in machine learning tasks, which we call the *barcode summary*.

## 3 EXPERIMENTAL DESIGN

In this section we overview details of the data we study, and outline our global and local studies using persistent homology.

### 3.1 DATA AND REPRESENTATIONS

We compute the barcodes of point clouds in $\mathbb{R}^D$, where each point corresponds to the latent representation of the last token of a given input in a given layer, or of a 2D embedding described in Section 3.3. The choice of the last token is justified by its role in encoding the model's aggregated interpretation of the input context (Zou et al., 2023).

**TaskTracker (Clean vs. Poisoned).** The TASKTRACKER (Abdelnabi et al., 2024) test set comprises $> 62$k examples, where each example contains the activation values for an input that is either *clean* or *poisoned*. Each input combines a user prompt with a retrieved data block. In poisoned cases, the data block contains an injected secondary task from several standard safety and security benchmarks, including BEAVERTRAILS (Ji et al., 2023), HARMBENCH (Mazeika et al., 2024), and JAILBREAK-BENCH (Chao et al., 2024). These attacks are applied across diverse task prompts, such as coding challenges and question-answering, to create realistic test cases. We analyze representation data from six instruction tuned LLMs; Phi3-mini-4k (3.8B parameters) and Phi3-medium-128k (14B parameters) (Abdin et al., 2024), Mistral 7B (Jiang et al., 2023), LLaMA3 8B and 70B (Grattafiori et al., 2024), and Mixtral-8x7B (Jiang et al., 2024).

**Sandbagged Models (Locked vs. Elicited).** We fine-tuned Mistral 7B and LLaMA3 8B on a corpus of 3,347 synthetic questions inspired by the WMDP benchmark (Li et al., 2024a) using LoRA (Hu et al., 2021). Following the methodology of van der Weij et al. (2024), we created models with two operational modes: *locked* (default state) and *elicited* (activated by a specific password in the prompt). We extracted final-token representations from each layer during inference with evaluation prompts in both states. Further implementation details are provided in Appendix E.

### 3.2 GLOBAL LAYER-WISE ANALYSIS

This analysis establishes and explains a consistent topological distinction between normal and adversarial representations, following the pipeline in Figure 3. We used RIPSER++ (Bauer, 2021; Zhang et al., 2020) to compute barcodes, leveraging subsampling techniques, both to reduce the computational cost of PH and to enable statistically robust inference. Subsampling approaches in PH are theoretically grounded, as under mild sampling models, persistence diagrams estimated from point clouds converge to the population diagrams with guaranteed rates (Chazal et al., 2015; 2014). For each model layer, we drew $K = 64$ subsamples of $k = 4096$ normal representations; and $K = 64$ subsamples of $k = 4096$ adversarial representations—see Appendix C.2 for ablations. We vectorized the corresponding barcodes into 41-dimensional barcode summaries (cf. Section 2.3), and performed the analysis in Figure 3, see results and further details in Section 4.1.

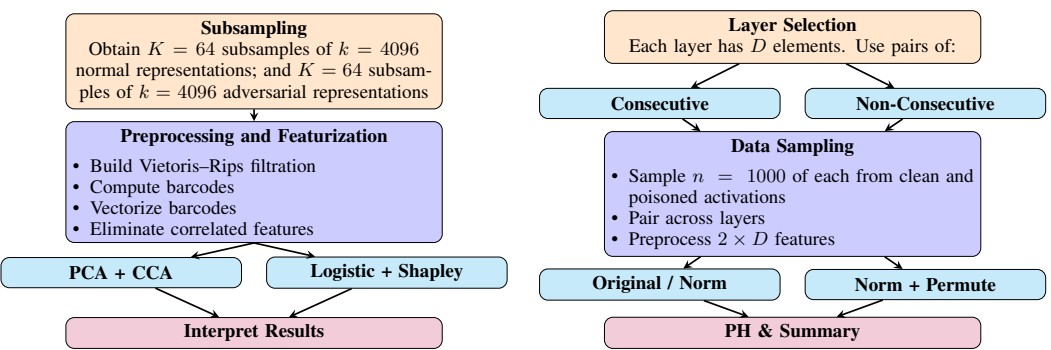

Figure 3: Pipeline for layer-wise topological analysis.

Figure 4: Pipeline for local analysis.

### 3.3 Local Information Flow Analysis

This analysis quantifies neuron-level information flow by tracking topological changes in activation patterns between layers. For each pair of layers $\ell$ and $\ell'$, we construct a 2D point cloud from their corresponding $D$-dimensional activation vectors. Each of the $D$ points in this embedding has coordinates $(v_i^\ell, v_i^{\ell'})$, representing the activation of the $i$th neuron in layer $\ell$ and layer $\ell'$, respectively.

The rationale for this embedding is that activations between consecutive layers are empirically highly correlated, causing points to cluster near the identity line $y = x$, as shown in Figure 5a. Significant transformations in network processing are reflected in neurons whose activations deviate from this line, producing topological structures (e.g., loops) that PH captures and quantifies. We apply this analysis to 1000 clean and 1000 adversarial activation samples to compare the resulting topological signatures, which are presented in Section 4.2.

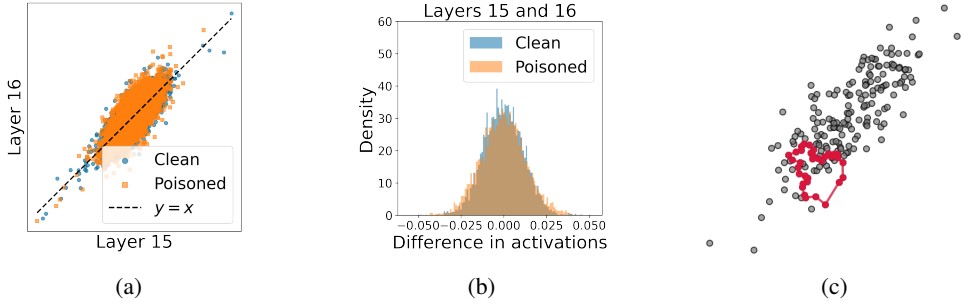

(a)  (b)  (c)

Figure 5: **(a):** Example 2D embedding showing correlation of activations in consecutive layers. **(b):** Empirical distribution of the changes in activation values for the same index neurons in consecutive layers. **(c):** Cycle corresponding to a long 1-bar in the PH barcode of the point cloud in (a).

## 4 Results

We now present the implementation results of our proposed analyses to the data described above.

### 4.1 Global Analysis: The Shape of Adversarial Influence

Our global analysis, as outlined in Figure 3, reveals a consistent and highly discriminative topological signature of adversarial influence across all six LLMs. Specifically, we show that adversarial inputs induce a "topological compression" of the latent space. Here, we present the results of quantifying and interpreting the effect of XPIA on Mistral 7B's latent space. Results for the other five models are provided in Appendix C.3. Results for the Mistral 7B and LLaMA3-7B models subjected to the backdoor finetuning attack for sandbagging are given in Appendix C.4.

**Cross-Correlation Analysis of Barcode Summaries.** In Figure 6, a growing block of highly correlated features appears in the cross-correlation matrix of the 41 features of the barcode summaries. To reduce redundancy and prevent overfitting, we removed highly correlated variables, ensuring an efficient and informative representation for more parsimonious models in subsequent analyses. We discarded all features that have a correlation higher than a threshold of 0.5 with at least one feature present in the analysis, resulting in the features in Table 6. We refer to this data set as the *pruned barcode summaries*. The first feature appearing in this block of highly correlated features is the mean deaths of the 0-bars (the average of their ending points), which is retained in the pruned barcode summaries as representative of the block. However, we remark that the prominence of this statistic in the results of our analysis does not imply a lack of significance for higher-order topological features (specifically, 1-bars). Empirically, there is a strong correlation between statistics of the 0- and 1-bars in our results; theoretically, it is known that the deaths of 0-bars are closely linked to the births of 1-bars (which has been explored using Morse theory; see Adler & Taylor (2011)). Having identified the most informative features, we next examine whether they support geometric separation in PCA space.

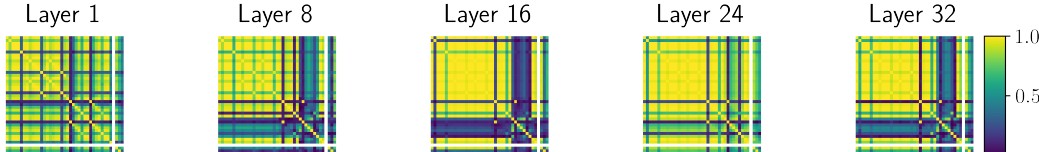

Figure 6: **Cross-correlation matrices for the barcode summaries** for clean vs. poisoned activations.

**Geometric Separation of Latent States.** The projection of the pruned barcode summaries over their first two principal components (Figure 7) clearly separates the normal and adversarial modes across layers. This is consistent with the intuition presented in Figure 1, where a single barcode of a clean sample with $n = 1000$ activations (corresponding to a point in the PCA plot) was visibly different than the barcode of a poisoned sample with same number of points. To better quantify and test the different topology between these modes, we used cross correlation analysis (CCA) to identify the topological features driving this separation. CCA works by quantifying linear relationships between two multivariate datasets by finding pairs of canonical variables with maximal correlation. The *loadings* are the contributions of individual features to these canonical variables, measuring their importance in capturing the relationship. We found that mean deaths of the 0-bars ranked first in all layers, and that the number of 1-bars appeared as a significant statistic as well (see Figure 18).

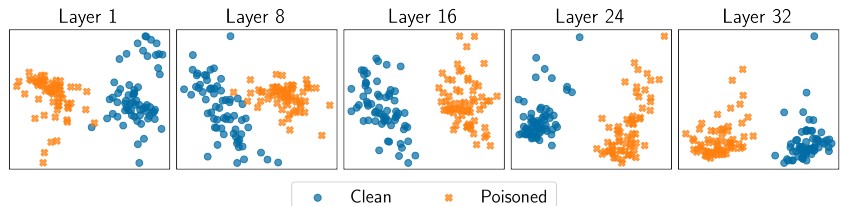

Figure 7: **PCA of pruned barcode summaries of clean vs. poisoned activations**. Clear distinction appears in the two first PC projections from the PCA of the pruned barcode summaries for layers 1, 8, 16, 24, and 32. The explained variances are 0.59, 0.49, 0.52, 0.96 and 0.83, respectively.

**Discriminative Power of Topological Features.** We trained a logistic regression model on the pruned barcode summaries to quantify how well the topological features alone distinguish between normal and adversarial inputs (subsamples). We obtained perfect accuracy and AUC–ROC on the test data, and 5-fold cross validation over the training data (Figure 8). As a baseline comparison, we trained a linear discriminant analysis (LDA), a linear support vector machine (SVM), and a logistic regression to distinguish 1000 clean and 1000 poisoned activations, raw and after reducing dimensionality using a sparse autoencoder (AE) with hidden dimension 128; see Table 1 for results. We found that the barcode summaries outperform these methods in general, particularly for early layers. However, we emphasize that the information that they encode must be understood as complementary

to that of the linear methods above, and that our true interest in the outstanding predictive power of barcode summaries resides in the fact that feature importance methods applied to the trained logistic regression allow us to interpret the differences in topology between clean and poisoned data, which is our ultimate goal.

Table 1: **Comparison of predictive power with linear methods**. Accuracy, with a 70/30 train/test split, of a linear discriminant analysis (LDA), a linear SVM and a logistic regression (LR) trained to distinguish 1000 raw clean activations from 1000 raw poisoned activations, with or without reducing the dimensionality of the data using a sparse autoencoder (SAE); and our method using PH.

| Layer | LDA | LDA (SAE) | SVM | SVM (SAE) | LR | LR (SAE) | PH |
|---|---|---|---|---|---|---|---|
| Layer 1 | 0.995 | 0.995 | 0.8875 | 0.7400 | 0.8700 | 0.7425 | 1.0000 |
| Layer 8 | 1.000 | 0.998 | 1.0000 | 0.6425 | 0.9950 | 0.6225 | 1.0000 |
| Layer 16 | 1.000 | 0.9975 | 1.0000 | 0.8125 | 1.0000 | 0.6725 | 1.0000 |
| Layer 24 | 1.000 | 0.9975 | 1.0000 | 0.9975 | 1.0000 | 0.9600 | 1.0000 |
| Layer 32 | 1.000 | 1.0000 | 1.0000 | 1.0000 | 1.0000 | 1.0000 | 1.0000 |

We used Shapley (or SHAP) values to interpret the model's performance. Shapley values quantify the contribution (with sign) of each feature to the prediction of the model for a given input. Our analysis revealed that the mean of 0-bar deaths and the number of 1-bars strongly influence predictions, exhibiting a clear dichotomous effect: points with smaller mean death in their 0-bars and bigger number of 1-bars are typically classified as clean, whereas points with bigger mean death of their 0-bars and smaller number of 1-bars are classified as poisoned.

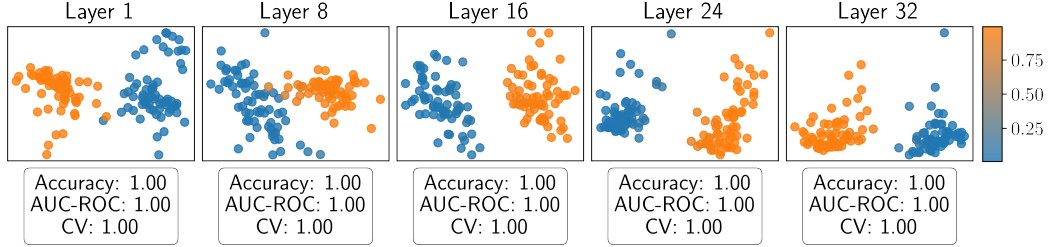

Figure 8: **Logistic regression for clean vs. poisoned activations** trained on a 70/30 train/test split of the pruned barcode summaries, plotted on the projection onto the two first PCs. Accuracy and AUC–ROC on the test data and 5-fold cross validation on train data are presented for each model.

**The Signature of Topological Compression.** Interpreting the distributions of the barcode summaries for clean vs. poisoned data reveals that adversarial conditions typically yield fewer 1-bars (loops) forming at later scales, yet persisting longer (see Figure 20). Conversely, the non-adversarial conditions tend to form earlier loops with more uniform lifetimes (higher persistent entropy). This pattern aligns with the Shapley value results (Figure 19): lower mean death times of 0-bars (i.e., more compact point clouds) are associated with predictions of "clean", while higher values (more spread-out clouds) shift predictions toward "poisoned". Similarly, a lower number of 1-bars tends to indicate "poisoned", whereas a higher count suggests "clean".

Thus, global topological features indicate a consistent distortion where adversarial states "compress" the representation space in a way that results in larger loops in fewer directions. In comparison, non-adversarial states have a larger number of smaller loops that are more evenly distributed and higher-entropy. This signature is robust, persisting even against adaptive attacks from the large-scale LLMail-Inject red teaming dataset that include attacks explicitly designed to evade the TaskTracker activation based defence (see Appendix G). A more detailed analysis across all models, layers, and adversarial conditions is provided in Appendix C and summarized in Table 2.

**Local Dispersion Ratio Across Poisoned Conditions.** To quantify how poisoning alters localized geometry in hidden-layer representation space, we use the *local dispersion ratio (LDR)*. For each final token's activation difference vector we identify its $k$ nearest neighbors in each layer and perform PCA on those points. Let $\lambda_1 \geq \cdots \geq \lambda_{D'}$ be the resulting eigenvalues, where $D' = \min(k, D)$,

Table 2: **Topological compression signature across models.** Adversarial inputs reshape latent space geometry by increasing the mean death time of connected components ($\bar{d}_{H_0}$ increases), reducing the number of loops ($\#H_1$ decreases), and extending the lifetime of remaining loops ($\bar{\ell}_{H_1}$ increases). Each cell indicates whether the adversarial condition matches this pattern ($\checkmark$), is inconsistent across layers ($\sim$), or shows the opposite ($\times$). Acc. is the minimum logistic-regression test accuracy across layers.

| Model | Acc. | $\bar{d}_{H_0}$ | $\#H_1$ | $\bar{\ell}_{H_1}$ |
|---|---|---|---|---|
| Phi3-mini (3.8B) | 1.00 | $\checkmark$ | $\checkmark^a$ | $\checkmark$ |
| Mistral (7B) | 1.00 | $\checkmark$ | $\checkmark$ | $\checkmark$ |
| LLaMA3 (8B) | 1.00 | $\checkmark$ | $\checkmark$ | $\checkmark$ |
| Mixtral-8×7B | 1.00 | $\checkmark$ | $\checkmark$ | $\checkmark^a$ |
| Phi3-medium (14B) | 1.00 | $\checkmark$ | $\checkmark$ | $\checkmark$ |
| LLaMA3 (70B) | 1.00 | $\checkmark$ | $\sim^b$ | $\checkmark$ |

[a] Holds at L1–L24 but inverts at L32.   [b] Direction varies across layers with no dominant trend.

since PCA on $k$ points in $D$ dimensions yields at most $\min(k, D)$ eigenvalues. The *dispersion ratio* is then defined as $\frac{\sum_{j=2}^{D'} \lambda_j}{\lambda_1 + \epsilon}$, where $\epsilon$ prevents division by zero. A higher LDR indicates that variance is more evenly spread among secondary directions, whereas a lower LDR implies most variance lies in a single dominant direction. Appendix B.3 further stratifies poisoned conditions into executed, refused, and ignored subclasses and shows that executed and ignored attacks exhibit elevated LDR in mid-layers relative to clean prompts. This indicates that the model allocates additional representational capacity to elaborating the injected instructions, whereas refused attacks are mapped into a more compressed, low-dispersion region, directly linking layer-wise geometric changes to task-level model behavior. Figure 9 shows that LDR differences remain tightly centered around zero under Clean vs. Clean and Poisoned vs. Poisoned resampling, confirming negligible within-class variability. In contrast, Mixed vs. Mixed splits exhibit systematic deviations that mirror the clean–poisoned separation observed in Figures 11 and 12 of Appendix B.3, indicating that LDR captures genuine geometric differences rather than artifacts of sampling noise or random partitioning.

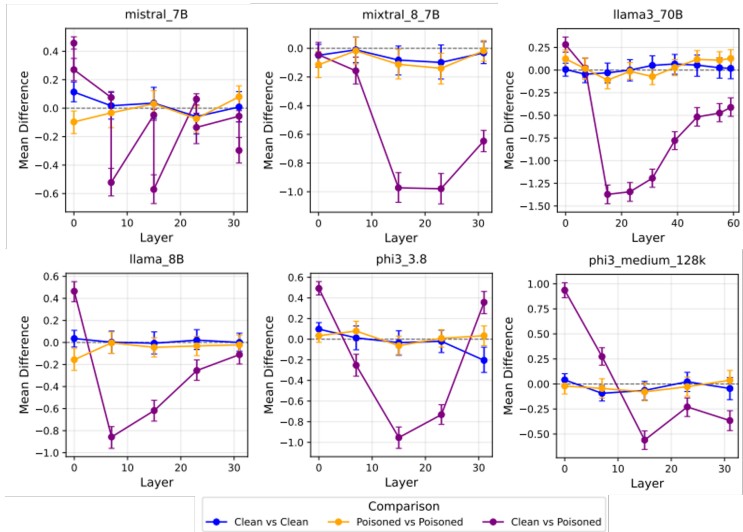

Figure 9: **Ablation of dispersion ratio differences (Clean vs. Clean, Poisoned vs. Poisoned, Mixed vs. Mixed).** Each plot shows the difference in mean dispersion ratio (clean minus poisoned). Positive values indicate that the clean subset exhibits higher dispersion, whereas negative values reflect a more dispersed poisoned subset.

## 4.2 LOCAL ANALYSIS: INFORMATION FLOW BETWEEN LAYERS

To investigate the fine-grained mechanisms of adversarial influence, our local analysis quantifies how information transforms between layers at the neuron level. We present the results for Mistral 7B below; see Appendix D.2 for other models.

**Analysis on Consecutive Layers.** Our local method revealed a structural phase shift in the network's information flow under adversarial influence. We computed Vietoris–Rips PH barcodes of the 2D embeddings described in Section 3.3 for the activations normalized to zero mean and unit variance. This is to ensure that any differences in topological features aren't due to scaling differences and to control the condition where neuron indices are randomly permuted. We measured the topological complexity by the total persistence of 1-bars, and found significant differences between clean and poisoned activations across layers in the raw and normalized activations (Figure 10 (left)). Furthermore, the ratio of topological complexity between clean and poisoned activations (Figure 10 (center)) shows that clean inputs initially exhibit a more complex structure that simplifies in deeper layers. In contrast, poisoned activations start simpler but their topological complexity increases, diverging significantly from the clean activations around layer 12. This suggests that adversarial influence causes a major reconfiguration of information processing in the model's deeper layers. The disappearance of this signal in the permuted control condition (shown in Figure 56 of the Appendix D.2.1) confirms that the effect relies on specific neuron-to-neuron pathways rather than arising from a statistical artifact.

Table 3: **Peak analysis.** Precision@$k$ for $k$=1, 3, and 5 largest peaks in total variance, and their precision in detecting the largest peaks in absolute difference between the two classes. Spearman's rank correlation ($r$) is reported in the last column. *, ** correspond to $p$-values $<.05$ and .01, respectively.

|                          | $p@1$ | $p@3$ | $p@5$ | $r$      |
|--------------------------|-------|-------|-------|----------|
| Total Persistence 0-bars | 0     | .33   | .4    | 0.46**   |
| Total Persistence 1-bars | 0     | .67*  | .8**  | 0.78**   |
| Mean Birth 1-bars        | 1.0*  | .33   | .8*   | 0.46**   |
| Mean Death 1-bars        | 1.0*  | .33*  | .8**  | 0.69**   |

In a real-world setting without labels, these informative layers can still be identified. We found that the overall variance of a topological feature across all samples strongly correlates with the magnitude of the clean-vs.-poisoned difference (Figure 10 (right)). As shown in Table 3, we evaluated the alignment between overall variance and class separation using precision at $k$ ($p@k$) and Spearman's rank correlation ($r$). To validate statistical significance against a random baseline, we generated empirical null distributions via random permutations, with significance levels indicated by asterisks. The high precision (particularly at $k = 5$) and moderate-to-strong correlations indicate that layers with the highest variance are reliable indicators of those with the largest class separation. This provides a practical, unsupervised signal for locating where adversarial effects are most prominent.

A further example of how different barcode summaries propagate across the layers can be found in Appendix D.2.1 for Mistral 7B, showing the patterns for the mean deaths of 0-bars.

**Analysis on Non-Consecutive Layers.** We expanded the previous analysis to activations from non-consecutive layers to show that in neighboring layers, the model operates on similar groups of neurons, leading to element-wise interactions that construct meaningful topological features distinguishing clean from poisoned datasets. The ratio of the mean death times of 0-bars between clean and poisoned activations as the layer interval increases is shown in Appendix D.2.5. For layer intervals of 1 and 3, the ratios for normalized activations and the control setting remained distinct, indicating meaningful topological interactions. However, at an interval of 10 layers, the scaled and control settings showed significant overlap, suggesting a much diminished difference in the interactions in clean and poisoned data. A similar pattern can be observed for other barcode summaries, such as the total persistence of 1-bars, see Appendix D.2.5.

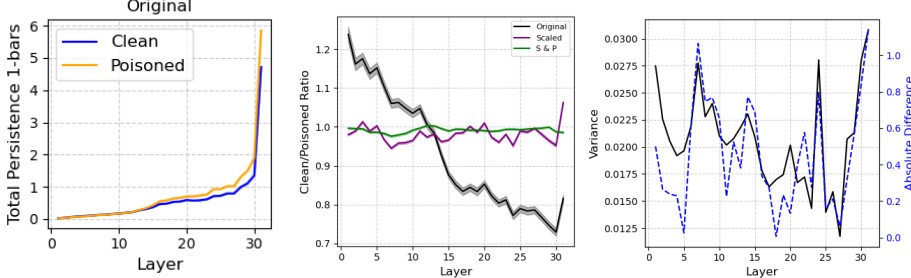

Figure 10: **Local analysis of consecutive layers for the total persistence of 1-bars.** Comparisons of the average total persistence of 1-bars across 1000 samples for Mistral model using original activation data **(left)**. **(center)** Ratios of mean total persistence of 1-bars between clean and poisoned datasets for original, scaled, and scaled and permuted activations. **(right)** Overlaid plots of the overall variance of total persistence of 1-bars for clean and poisoned datasets combined and the absolute difference between mean total persistence of 1-bars for clean and poisoned datasets.

## 5 DISCUSSION AND FUTURE WORK

Our global and local analyses provide converging evidence that adversarial influence manifests as topological compression of an LLM's latent space, shifting representations from compact, diverse structures to more dispersed, topologically simpler ones. This signature is consistent across architectures (3.8B–70B parameters), attack vectors (prompt injection and backdoor fine-tuning), and layers. This topological approach offers a distinct and complementary form of interpretability that is relational rather than compositional. While methods such as sparse autoencoders (SAEs) (Cunningham et al., 2023) decompose representations into interpretable "building block" features but analyze each activation in isolation, making them blind to the nonlinear geometry that emerges from interactions between activations. Moreover, because SAE dictionaries are tied to specific model weights, they cannot be reliably compared across models or fine-tuning stages. Our PH-based framework sidesteps both limitations by computing intrinsic, coordinate-free geometric properties that are directly comparable across architectures and training conditions.

The implications of our work extend to the core of interpretability and AI safety. Our findings contribute to a growing body of evidence that a model's behaviors are encoded in the geometry of its latent space. This perspective aligns with work showing that memorization corresponds to a reduction in the effective dimensionality of the representation manifold (Stephenson et al., 2021), and that the success of linear probes may stem from their ability to approximate more complex topological structures (Engels et al., 2025). Our finding that adversarial influence induces a "topological compression" provides new evidence for this view, whereby the model's transition to an out-of-distribution operating mode is accompanied by a collapse in geometric complexity. More broadly, our work suggests that safety-relevant properties like robustness to adversarial manipulation are not merely abstract behavioural outcomes but quantifiable geometric characteristics of the representation space itself.

**Limitations.** We do not attempt to interpret the semantic content of the cycles and topological features we identify. Furthermore, we compute Vietoris–Rips filtrations only in dimensions 0 and 1. While this proved more than sufficient for our analysis, computing higher-dimensional homology could yield additional insights, provided one can manage the quadratic scaling of Vietoris–Rips computations with respect to the number of data points.

**Future Work.** Our work only scratches the surface of the potential of PH applied to LLM interpretability. Future research might adapt classical PH techniques to account for specific architectural features of LLMs, with the goal of producing more interpretable features that map to semantic content. Another natural question is whether the topological compression phenomenon we observe is a general property of model misalignment and adversarial attacks (Stephenson et al., 2021). Finally, further investigation is needed into how such topological awareness might be leveraged during model training and architecture design (Brüel-Gabrielsson et al., 2020).

ACKNOWLEDGMENTS

A.F. would like to thank Daniel Jones for productive discussions about the project. I.G.R. was funded by a London School of Geometry and Number Theory–Imperial College London PhD studentship at the time of research, which was supported by the EPSRC grant No. EP/S021590/1. Q.W. was funded by a CRUK–Imperial College London Convergence Science PhD studentship at the time of research, which was supported by Cancer Research UK under grant reference CANTAC721\10021 (PIs Monod/Williams). H.D. is supported by the research program "Change is Key!" supported by Riksbankens Jubileumsfond (under reference number M21-0021). H.D., A.M., and Q.W. are supported by the EPSRC AI Hub on Mathematical Foundations of Intelligence: An "Erlangen Programme" for AI No. EP/Y028872/1.

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

## A  PERSISTENT HOMOLOGY

We provide additional background on PH and the underlying mathematical formulation that supports its application as a tool to detect the *multiscale topological features* within data.

## A.1 THEORETICAL BACKGROUND

PH refers to a set of methods that are implemented to extract the shape and size of data a multiple scales. We now present the underlying mathematical principles that support this tool.

**Input data.** PH accommodates for diverse data modalities: images, point clouds, graphs, etc. One of the most basic yet general data types that it accepts is *finite metric spaces*, i.e., finite subsets $S \subset X$ of some metric space $(X, d)$. Restricting $d$ to $S$, we obtain a notion of dissimilarity between the points in our metric space. This is the data modality that we will consider for the remainder of the section, as it encompasses most of the real data that we encounter.

**Filtrations.** The first step in the PH pipeline consists of constructing a filtration from our input data, that is, a family of nested topological spaces. For computational and storage reasons, *simplicial complexes* are often favored as the topological spaces appearing in the filtration. An abstract simplicial complex $K$ over a vertex set $S$ is defined as a set of subsets of $S$ which is closed under inclusion, i.e., if $\sigma \in K$ and $\tau \subset \sigma$, then $\tau \in \sigma$. Subsets $\sigma = \{s_{i_0}, \dots, s_{i_p}\}$ of $p+1$ elements are called $p$-simplices. There are various ways of defining a simplicial complexes from a discrete set $S$, and they usually depend on fixing a scale parameter $\epsilon > 0$.

For instance, in this work, we have leveraged the *Vietoris–Rips complex*, obtained by considering all the subsets $\sigma$ of $S$ with $\mathrm{diam}(\sigma) := \max_{s,s' \in \sigma} d(s, s')$ less or equal than $\epsilon$,

$$\mathrm{VR}_\epsilon(S, d) := \{\emptyset \neq \sigma \subset K : \mathrm{diam}(\sigma) \leq \epsilon\}. \tag{1}$$

The implementation of this complex is straightforward, and has the advantage that it is only necessary to store the pairwise distance between points in $S$ to build it. However, it has the disadvantage of exploding in size with the number of points: if $S$ has $n$ points, then $|\mathrm{VR}_\epsilon(S, d)| = O(2^n)$ (see Table 1 in Otter et al. (2017))

An alternative is the *Čech complex* at scale $\epsilon \geq 0$, where a simplex $\sigma = \{s_{i_0}, \dots, s_{i_p}\}$ belongs to the complex if and only if all the balls of radius $\epsilon$ centered at the points of the simplex have nonempty intersection,

$$\check{\mathrm{C}}_\epsilon(S, d) := \left\{ \emptyset \neq \sigma \subset S : \bigcap_{s \in \sigma} B(s, \epsilon) \neq \emptyset \right\}. \tag{2}$$

The Čech complex has very nice theoretical properties (for instance, it satisfies the conditions of the Nerve Theorem). However, it has similar complexity to the Vietoris–Rips complex, and in fact we have

$$\check{\mathrm{C}}_\epsilon(S, d) \subseteq \mathrm{VR}_\epsilon(S, d) \subseteq \check{\mathrm{C}}_{\sqrt{2}\epsilon}(S, d).$$

A final option to consider, which significantly reduces the number of simplices in the complex, is the *alpha complex*. To make this simplicial complex coarser, the idea is to intersect the balls centered around the points in the point cloud, $B(s, \epsilon)$, with their Voronoi cells, $V(s)$, and thus define $R(s, \epsilon) := B(s, \epsilon) \cap V(s)$. The Voronoi cells form a partition of the metric space $X$ where the points in each region are closest to the same point in $S$. Since both $B(s, \epsilon)$ and $V(S)$ are convex, their intersection $R(s, \epsilon)$ remains convex. From the definition of the Voronoi cells, these spaces $R(s, \epsilon)$ are either disjoint or overlap along their boundary, significantly reducing the number of intersections between them. The alpha complex is thus defined as

$$\alpha(S, \epsilon) := \left\{ \emptyset \neq \sigma \subset S : \bigcap_{s \in \sigma} R(s, \epsilon) \neq \emptyset \right\} \tag{3}$$

and is significantly smaller in size due to the introduction of the Voronoi cells.

The Vietoris–Rips, Čech, and alpha filtrations are defined considering the families of the corresponding complexes for all values of the parameter $\epsilon \geq 0$. Since the conditions for including simplices are relaxed as $\epsilon$ increases, we obtain the defining condition of a filtration $\{K_\epsilon : \epsilon \geq 0\}$, namely that for $\epsilon \leq \epsilon'$ we have $K_\epsilon \subset K_{\epsilon'}$. There are additional types of filtrations that we do not cover here, such as cubical filtrations (particularly suited for images) or witness complexes (based on having some landmarks or witnesses in our point cloud). We refer to Otter et al. (2017) for a survey and further details on these constructions.

**Homology and persistence modules.** Leveraging tools from algebraic topology, we can compute the *simplicial homology groups* associated to a given simplicial complex $K$, which come in various degrees $H_p(K)$, for $p \geq 0$ an integer number, and are topological invariants of the complex. They contain information about its topological features, for $p = 0$ these correspond to components or clusters, for $p = 1$ to loops or holes, for $p = 2$, to bubbles or cavities, and so on for higher values of $p$. The homology construction is functorial, meaning that there is an assignment which for a map $f : K \to K'$ between two simplicial complexes, provides a linear map at the homology level $H_p(f) : H_p(K) \to H_p(K')$, preserving the identity and composition. Applying this to any of the filtrations of the step above we obtain a *persistence module*, that is, a family of vector spaces $\{H_p(K_\epsilon) : \epsilon \geq 0\}$ endowed with linear maps $H_p(\epsilon \leq \epsilon') : H_p(K_\epsilon) \to H_p(K_{\epsilon'})$ for $\epsilon \leq \epsilon'$, which are the maps induced by the inclusions of the filtration. In other words, $H_p(K_\bullet)$ can be seen as a functor from the poset category $(\mathbb{R}_{\geq 0}, \leq)$ to the category of vector spaces and linear maps. Given the mathematical construction of homology, $H_p(K_\bullet)$ contains information about the topological features in the simplicial complexes of the filtration, and in particular, about when features appear and disappear as the parameter $\epsilon$ increases. We now seek to provide a compact description for this.

**Persistence barcodes.** The mathematical structure of a persistence module has various desirable properties. Among them, one of the most important ones is satisfying the conditions for the the so called *structure theorem* (Botnan & Lesnick, 2023, Theorem 4.2) to apply, which tells us that a given a persistence module $H_p(K_\bullet)$ decomposes in an essentially unique way as a direct sum of interval modules $\mathbb{R}[b, d)$. Interval modules are persistence modules supported over intervals of the real line which, inside their support, map to the vector space $\mathbb{R}$, and outside, to $0$. Since the decomposition is an invariant of the isomorphism type of $H_p(K_\bullet)$, the collection of intervals appearing in it is also a topological invariant. We refer to this collection of bars as the *persistence barcode* of the input data. The interpretation of these barcodes becomes apparent: each of the bars in the barcode correspond to a topological feature that appears at the initial point in the interval (its *birth time*) and persists until its end (its *death time*). There are many other invariants that we can derive from the original persistence module $H_p(K_\bullet)$, such as the rank function (Frosini, 1990; 1992), the persistence image (Adams et al., 2017) or the persistence landscape (Bubenik, 2020); some of these invariants act on barcodes as vectorizations or embeddings. In this work, we focus on barcodes and we represent statistics calculated from bars and barcodes in the form of a vector, which is different in spirit from an embedding or vectorization of a barcode.

### A.2 PERSISTENT HOMOLOGY BARCODE STATISTICS

To interpret the barcodes from Section 3.2 and Section A.1, we extract key summary statistics that quantify the topological structure observed at each layer under both adversarial conditions.

From each 1-dimensional (1D) barcode, we gather intervals $(b_i, d_i)$ with $d_i > b_i > 0$ and define $\ell_i = d_i - b_i$. Forming a discrete distribution $p_i = \ell_i / \sum_j \ell_j$, the *persistence entropy* is

$$E = -\sum_i p_i \ln(p_i + \epsilon),$$

where $\epsilon$ is a small positive constant (e.g., $10^{-12}$) to ensure numerical stability. Higher $E$ indicates a more uniform distribution of lifetimes (no single interval dominates), whereas lower $E$ reflects a small number of long-lived intervals.

In addition to **entropy**, we compute the following summary statistics on dimension-1 bars:

- **Mean births (1-bars):** Average birth time $\bar{b}$
- **Mean deaths (1-bars):** Average death time $\bar{d}$
- **Mean persistence (1-bars):** Average lifetime $\overline{(d_i - b_i)}$
- **Number of 1-bars:** Count of finite intervals in dimension 1

We perform these computations for each barcode individually and then average over all barcodes in the same condition (elicited or elicited) and (clean or poisoned).

# B  FURTHER TOPOLOGICAL AND LOCAL VARIANCE INTERPRETATION

## B.1  EXTENDED PROMPT INJECTION (CLEAN VS. POISONED)

Table 4: **Dimension-1 persistent homology differences (clean − poisoned) in key metrics for three models across several layers.** Positive values mean the clean condition has a higher value, while negative indicates poisoned is higher for that metric. All entries rounded to four decimals.

| Model | Layer | Mean births 1-bars_diff | Mean deaths 1-bars_diff | Mean persistence 1-bars_diff | Entropy 1-bars_diff | Number 1-bars_diff |
|---|---|---|---|---|---|---|
| LLaMA-3 (8B) | 1 | -0.0005 | -0.0006 | -0.0001 | 0.1665 | 86.9700 |
| | 8 | -0.0609 | -0.0608 | 0.0001 | 0.1213 | 79.5600 |
| | 16 | -0.3166 | -0.3249 | -0.0082 | 0.0188 | 17.9367 |
| | 24 | -0.9932 | -1.0256 | -0.0324 | 0.1595 | 80.0833 |
| | 32 | -18.3367 | -18.9290 | -0.5923 | 0.3348 | 192.4900 |
| Mistral (7B) | 1 | 0.0004 | 0.0004 | 0.0000 | 0.0172 | 3.7967 |
| | 8 | -0.0293 | -0.0295 | -0.0002 | 0.1485 | 118.9167 |
| | 16 | -0.2375 | -0.2421 | -0.0047 | 0.1938 | 154.7633 |
| | 24 | -0.5694 | -0.5815 | -0.0120 | 0.2070 | 153.9633 |
| | 32 | -14.7376 | -15.0558 | -0.3182 | 0.2239 | 166.4267 |
| Phi 3 (3.8B) | 1 | 0.0011 | 0.0009 | -0.0002 | 0.0101 | 4.3200 |
| | 8 | -0.4522 | -0.4675 | -0.0153 | 0.0888 | 59.0967 |
| | 16 | -1.7825 | -1.8293 | -0.0467 | 0.0696 | 48.7367 |
| | 24 | -6.2017 | -6.3448 | -0.1431 | 0.1665 | 116.9433 |
| | 32 | -1.0092 | -1.0503 | -0.0411 | -0.4857 | -136.1200 |

For mean births and mean deaths, all layers except layer 1 across models have negative differences, indicating that poisoned intervals emerge and die later in the filtration. The mean persistence is also higher for poisoned examples, especially in the middle layers. The entropy differences are also almost all positive (except Phi3 3.8's final layer), suggesting that clean distributes its 1D lifetimes more evenly, while poisoned tends to concentrate total persistence in fewer cycles. Likewise, the number of 1-bars remains mostly positive, so clean produces more 1D holes overall.

Overall, poisoned conditions often yield fewer but more dominant dimension-1 features that appear later and persist longer, whereas clean spreads its total persistence across numerous shorter-lived cycles, resulting in higher entropy. Depending on the model, the specific layer where these features distinctions appear strongest differ but the nature of the difference is the same across models and layers.

## B.2  SANDBAGGED MODEL (LOCKED VS. ELICITED)

Table 5: **Dimension-1 persistent homology differences (elicited − locked) for two models across multiple layers.** Positive values indicate that the elicited condition has higher values; negative means locked is higher for that metric.

| Model | Layer | Mean births 1-bars_diff | Mean deaths 1-bars_diff | Mean persistence 1-bars_diff | Entropy 1-bars_diff | Number 1-bars_diff |
|---|---|---|---|---|---|---|
| LLaMA-3 (8B) | 0 | -0.0127 | -0.0132 | -0.0005 | 0.0156 | 3.2400 |
| | 7 | -0.3425 | -0.3555 | -0.0130 | 0.0647 | 27.8600 |
| | 15 | -0.0476 | -0.0455 | 0.0021 | 0.2114 | 135.2900 |
| | 23 | -0.1168 | -0.1204 | -0.0037 | 0.0100 | 61.8766 |
| | 31 | -0.9750 | -1.0458 | -0.0707 | 0.0620 | 28.2800 |
| Mistral (7B) | 0 | -0.0053 | -0.0055 | -0.0002 | 0.0942 | 27.1533 |
| | 7 | -0.1925 | -0.1989 | -0.0064 | 0.0310 | 14.1066 |
| | 15 | 0.0393 | 0.0352 | -0.0041 | 0.0277 | 10.9300 |
| | 23 | 0.6722 | 0.7037 | 0.0315 | -0.0363 | -0.1900 |
| | 31 | 14.6450 | 15.2952 | 0.6503 | -0.0014 | 9.3233 |

For LLaMA3 8B , the mean birth and death differences are negative across all computed hidden layers (1, 8, 16, 24, 32). Note that layers are zero-indexed, meaning that layer 0 corresponds to the first hidden layer, layer 1. This indicates that, in the locked condition, 1D cycles exhibit larger (i.e., later) birth and death times compared to elicited. In other words, when locked, the 1D features tend to emerge "further out" in the filtration. The mean persistence difference between conditions is also negative (except layer 16), suggesting that locked cycles generally persist slightly longer on average. Entropy differences are positive, indicating that elicited exhibits a greater diversity or spread among the lifetimes of its 1D features. The number of 1-bars is positive (sometimes strongly so), meaning there are substantially more 1D features in the elicited condition.

We see similar results for Mistral 7B  with negative differences in births and deaths in earlier layers, implying that locked has larger birth/death times at those lower layers. However, the sign flips, with elicited displaying larger values for births, deaths, and persistence. Specifically, layer 32 shows a notably large positive difference (e.g., $+14.64$ for births, $+15.29$ for deaths), indicating that the final layer in elicited captures significantly later 1D cycles relative to locked. The number of 1-bars also tends to be higher in elicited at most layers, except for a minor negative at layer 23, again suggesting that elicited reveals a greater number of dimension-1 features.

## B.3 LOCAL DISPERSION RATIO ACROSS POISONED CONDITIONS

We analyze how local geometry in hidden-layer representation space differs between clean and multiple poisoned modes in six LLMs. We further classify poisoned prompts into three sub-types:

1. **Executed:** The injected request is recognized and carried out (indirect prompt injection).
2. **Refused:** The model identifies the injected content as malicious and issues a refusal, effectively "shutting down" any detailed elaboration.
3. **Ignored:** The model neither executes nor refuses, but effectively overlooks the injected prompt, proceeding as if it were absent.

For each final token's activation difference vector $\Delta\mathrm{Act}_\ell(x_i) \in \mathbb{R}^D$, we identify its $k$ nearest neighbors in layer $\ell$ and perform PCA on those points. Let $\lambda_1 \geq \cdots \geq \lambda_{D'}$ be the resulting eigenvalues. We define the *dispersion ratio* of $\Delta\mathrm{Act}_\ell(x_i)$ as

$$\frac{\sum_{j=2}^{D'} \lambda_j}{\lambda_1 + \epsilon},$$

where $\epsilon$ prevents division by zero. A higher ratio indicates that variance is more evenly spread among secondary directions, whereas a lower ratio implies most variance lies in a single dominant direction.

**Ablation: Clean vs. Clean, Poisoned vs. Poisoned, and Mixed.**   To confirm that dispersion discrepancies primarily reflect true clean vs. poisoned distinctions rather than random partitioning or mixture effects, we performed three auxiliary comparisons:

1. **Clean vs. Clean**: Split the clean set into two subsets, ensuring no significant difference arises from sampling within the same class.
2. **Poisoned vs. Poisoned**: Applied the same procedure to poisoned data to assess within-class variability.
3. **Mixed vs. Mixed**: Randomly partitioned a combined pool of clean and poisoned samples into two balanced groups.

**Note on Statistical Methods:** For every layer in each subplot, we computed the dispersion ratio for both clean and the specified poisoned (or refused, executed, ignored) samples. We then conducted a Welch's $t$-test on these two groups (clean vs. poisoned/other), applying false-discovery rate (FDR) correction across layers. We also verified approximate normality via kernel density estimates (KDEs) for each groups. Plot markers with stars indicate layers where $p_{\mathrm{FDR}} < 0.05$, confirming a

statistically significant difference in dispersion ratio. To select $k = 30$, we tested candidate neighborhood sizes across layers and models, measuring which $k$ produced the largest absolute difference in mean local dispersion ratio between clean and poisoned conditions.

### B.3.1 DISCUSSION OF RESULTS

Figures 11 and 12 highlight that:

- **Early Layers (Layer 1–8):** Across all poisoning modes, the clean condition consistently shows a higher dispersion ratio, suggesting that the model initially allocates broader representational capacity for normal inputs.
- **Mid Layers (Layer 16):** This pattern often flips, with poisoned prompts (especially executed or ignored) exceeding the clean baseline, indicating the network is dedicating extra directions to elaborate or "embrace" these injected requests. Conversely, refused prompts typically exhibit reduced dispersion, mapping disallowed content into a lower-variance region.

Interestingly, our findings align with the results of Stephenson et al. (2021), which indicate that memorization tends to emerge in deeper layers where the effective dimensionality shrinks. Consistent with that view, we observe that executed or ignored prompts show a higher dispersion in mid-layers, implying the model invests additional capacity there for those injected instructions. Meanwhile, a refused request is routed into a more compressed region, effectively "shutting down" further representational expansion. In this sense, deeper layers may provide a setting where the network can more sharply discriminate or overfit certain inputs—supporting the idea that final layers reflect a gradually compressed, yet strategically focused representation space.

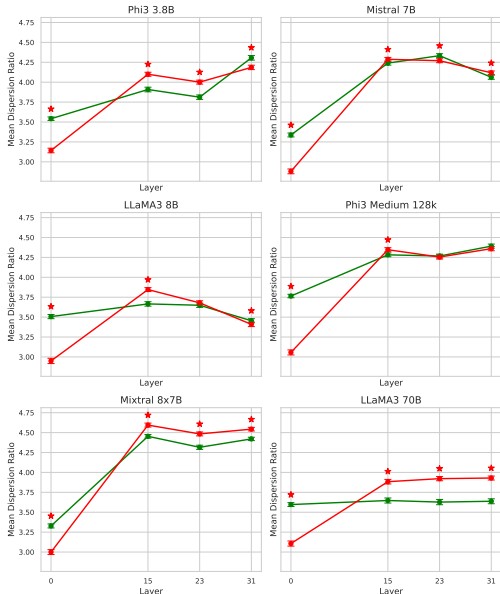

Figure 11: **Layer-wise Dispersion Ratio for Clean vs. Poisoned Examples.** The green and red lines depict mean dispersion ratios for clean and poisoned inputs, respectively, at different layer depths. Error bars around each point represent $\pm 1$ standard error of the mean (SEM). In early layers (left side), clean data consistently has higher dispersion on average, whereas in mid-layers (center), poisoned surpasses the clean baseline, indicating a re-distribution of representational capacity for the injected prompts. Layers where the difference is statistically significant ($p_{\text{FDR}} < 0.05$) are marked with a red asterisk above the higher mean value.

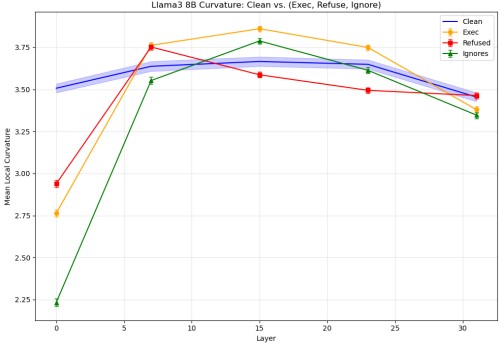

Figure 12: *LLaMA3_7B* **Dispersion Ratio: Clean vs. Executed, Refused, and Ignored Prompts.** The horizontal axis indicates layer depth, while the vertical axis represents the mean dispersion ratio. The blue curve (with confidence band) corresponds to clean inputs; orange, red, and green curves denote executed, refused, and ignored poisoned prompts, respectively. Notably, refused prompts show an early jump but then collapse below the clean baseline, whereas executed and ignored surpass it around mid-layers, highlighting distinct representational regimes.

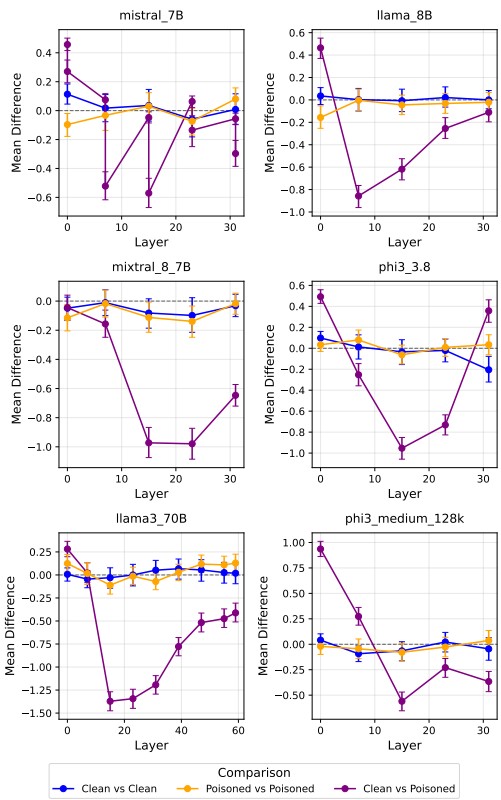

Figure 13: **Ablation of Dispersion Ratio Differences (Clean vs. Clean, Poisoned vs. Poisoned, Mixed vs. Mixed).** Each plot shows the difference in mean dispersion ratio (clean minus poisoned). Positive values indicate that the clean subset exhibits higher dispersion, whereas negative values reflect a more dispersed poisoned subset.

### B.4    COSINE DISTANCE OF REPRESENTATIONS

We analyze the difference representations $\Delta\mathrm{Act}_\ell(x_i) \in \mathbb{R}^D$ for corresponding pairs of clean and poisoned inputs in Figure 14. Specifically, for each model and layer, we load up to five pairs of clean and poisoned activation files, compute the difference between the activations for each pair, and con-

catenate these differences. From these differences, we draw equal-size subsamples of 5000 vectors. For each layer and comparison condition, we compute the mean pairwise cosine distance within each subsample. Because cosine distance is scale-invariant, we do not normalize these difference representations. We perform four comparison conditions: clean vs. poisoned, clean vs. clean (where clean samples are split in half), poisoned vs. poisoned (where poisoned samples are split in half), and mixed vs. mixed (where two separate mixed subsamples are created, each containing half clean and half poisoned differences). For each comparison, we generate two distributions of mean pairwise intra-class distances (or inter-class in the clean vs poisoned case) using 3 bootstrap iterations. We then apply Welch's $t$-test to these distributions to assess whether they diverge significantly.

Empirically, poisoned difference representations typically exhibit a higher mean cosine distance in deeper layers, indicating a more "spread-out" or heterogeneous arrangement of their difference vectors, much as we observed in the curvature analysis. clean data, by contrast, remains comparatively tightly clustered, implying less dispersion in its difference space. Interestingly, *LLaMA3_70B* displays similar characteristics in the early and final layers but poisoned representations have a noticeable smaller cosine distance in middle layers. This may reflect the ability of larger architectures to better partition representation space across the network before re-expanding in later layers.

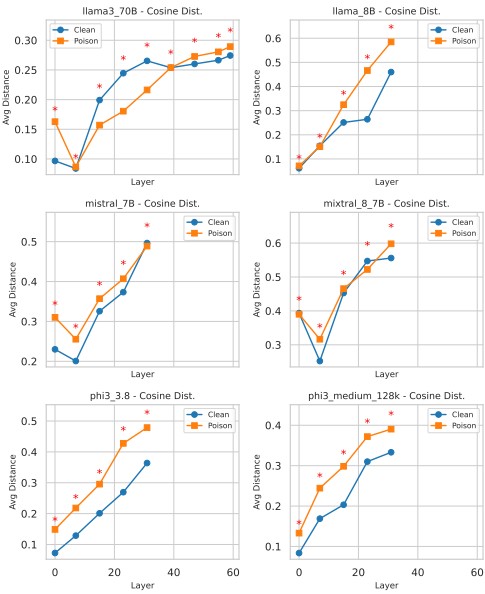

Figure 14: **Cosine Distance of Difference Representations Across Layers.** Each panel shows mean within-class distances (clean vs. poisoned) for the difference representations (*poisoned/clean pass* minus *baseline*), where higher values reflect greater variation among samples. Stars denote layers with significant differences.

## C   FURTHER DETAILS OF GLOBAL LAYER-WISE ANALYSIS

We now provide further details on the global layer-wise analysis.

### C.1   PIPELINE

We describe in more detail the pipeline in Figure 3 in the main text. Recall that our aim here was showcasing that topological signatures effectively capture distinctions between representations under normal or adversarial conditions, and to provide an interpretation of the reason behind such difference in terms of the "shape" of the latent representations.

We use RIPSER Bauer (2021) to compute barcodes, which is based on Vietoris–Rips filtrations (see Figure 2.1). The computational constraints of PH make it impossible to compute the barcode of

any of our two datasets (clean vs. poisoned or locked vs. elicited). Therefore, we leverage sub-sampling approaches (e.g., Chazal et al. (2015)) and compute barcodes from $K = 64$ subsamples $\{x_{i_1,\ell}, \ldots, x_{i_k,\ell}\} \subset \mathbb{R}^D$ with size $k = 4096$, of the representations per layer $1 \leq \ell \leq L$. From these, 64 are taken from normal activations and 64 from adversarial activations. We use these as proxies for the topology of the whole space.

Following Ali et al. (2023), we represent these barcodes as 41-dimensional feature vectors, which we call *barcode summaries*. These include 35 statistics derived from a $7 \times 5$ grid of {mean, minimum, first quartile, median, third quartile, maximum, standard deviation} $\times$ {death of 0-bars, birth of 1-bars, death of 1-bars, persistence of 1-bars, ratio birth/death of 1-bars}; as well as the total persistence (i.e., sum of the lengths of all bars in the barcode), number of bars, and persistent entropy (Chintakunta et al., 2015; Rucco et al., 2016) defined in Appendix A.2 for 0- and 1-bars. We reduce the dimensionality case-by-case, by eliminating highly correlated features (above a threshold of 0.5) through cross-correlation analysis.

For exploratory analysis, we apply PCA and compute CCA loadings to measure feature correlations with the principal components. A logistic regression model is then used for classification, and Shapley values (Lipovetsky & Conklin, 2001) are computed to evaluate feature importance. Shapley values, derived from cooperative game theory, quantify the contribution of each feature to model predictions by measuring its influence in shifting predictions from a baseline (e.g., 0.5 for logistic regression), providing an interpretable, feature-level analysis of predictive impact.

## C.2 ABLATION STUDIES ON SUBSAMPLING PARAMETERS

We evaluate the representation of clean and poisoned activations using a subsampling-based topological analysis. For each experiment, we consider a fixed layer of Mistral 7B and draw $k$ subsamples of size $n$ from the clean activations and $k$ subsamples of size $n$ from the poisoned activations. Each subsample is used to compute a Vietoris–Rips persistence diagram, which is subsequently represented as a 41-dimensional barcode summary vector. This procedure produces a combined point cloud in $\mathbb{R}^{41}$ of size $2k$, consisting of $k$ clean and $k$ poisoned feature vectors.

**Predictive Power of Barcode Summaries for Varying** $(n, k)$**.** We perform the same classification task as in the main text, namely, we fit a logistic regression model to classify between clean and poisoned in each point cloud with fixed $(n, k)$, for the first, the middle, and the last layer of Mistral 7B. We report the 5-fold cross validation results in Figure 15. We observe that there are no clear dependencies of this parameter over the parameters $(n, k)$. Layer 1 seems to be more difficult to classify, requiring at least 500 subsamples, whereas for later layers we obtain perfect classification with as little as $k = 30$ subsamples of size $n = 100$.

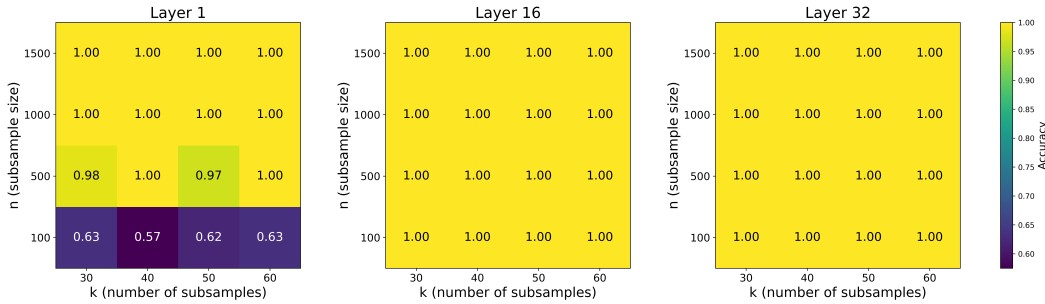

Figure 15: Accuracies of 5-fold cross validation on a logistic regression trained to distinguish barcode summaries of $k$ subsamples of size $n$ of clean activations and $k$ subsamples of size $n$ of poisoned activations at layers 1, 16 and 32 of Mistral 7B.

**Metric Description of Clusters for Varying** $(n, k)$**.** We now focus on activation values for layer 16 in Mistral 7B, over which the barcode summaries are computed in subsamples with parameters $(n, k)$. All feature vectors are standardized using a global `StandardScaler` fitted on the whole point cloud. We then compute several metrics to quantify the structure of the resulting represen-

tation: (i) the mean intra-class distance within the clean and poisoned subsamples, (ii) the mean inter-class distance between the two groups, and (iii) the inter-to-intra distance ratio

$$r := \frac{d_{\text{inter}}}{\frac{1}{2}\left(d_{\text{intra}}^{\text{clean}} + d_{\text{intra}}^{\text{poison}}\right)}.\qquad(4)$$

We perform ablations over the subsample size $n$ and the number of subsamples $k$. The intra-class distances (Figure 16 left and center) show minimal dependence on $k$, but decrease consistently as $n$ increases. This suggests that the barcode representations become more concentrated when subsamples contain more points. The values for $n = 500$, 1000, and 1500 are in close proximity, indicating an early convergence of this statistic with respect to $n$.

The inter-class distance (Figure 16 right) exhibits a complementary trend: it is largely invariant under changes in $k$, but increases with $n$. As before, the curves for $n = 1000$ and $n = 1500$ almost coincide, further supporting a convergence regime at moderate subsample sizes.

To combine these effects, we evaluate the inter-to-intra distance ratio in Figure fig. 17. This ratio remains stable across values of $k$, but increases with $n$, indicating that the relative separation between clean and poisoned representations improves as subsample size grows. The near overlap of the values for $n = 1000$ and $n = 1500$ again suggests convergence in this regime, which supports the choice of subsample sizes used in the main experiments.

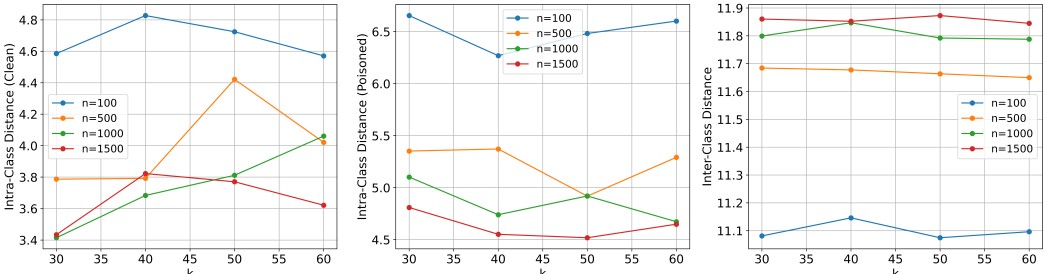

Figure 16: **Left:** Intra-class distance among the barcode summaries of $k$ subsamples of size $n$ of clean activations from layer 16 of Mistral 7B. **Center**: Intra-class distance among the barcode summaries $k$ subsamples of size $n$ of poisoned activations from layer 16 of Mistral 7B. **Right:** Intra-class distance among the clusters of clean and poisoned barcode summaries of $k$ subsamples of size $n$.

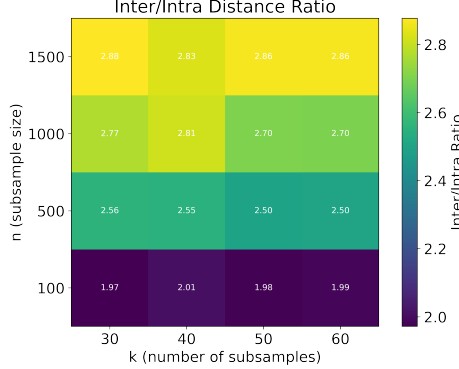

Figure 17: Inter-to-intra distance ratio (Equation 4 between $k$ subsamples of $n$ clean activations and $k$ subsamples of $n$ poisoned activations in layer 16 of Mistral 7B.

## C.3 RESULTS: CLEAN VS. POISONED

### C.3.1 MISTRAL 7B

We present here additional results on the global analysis for Mistral 7B that are referred to in the main text.

Table 6: **Pruned barcode summaries for layers 1, 8, 16, 24 and 32.** Features from the barcode summaries with correlation less than 0.5 in the cross-correlation matrix.

|  | Layer 1 | Layer 8 | Layer 16 | Layer 24 | Layer 32 |
|---|---|---|---|---|---|
| Mean death 0-bars | ✓ | ✓ | ✓ | ✓ | ✓ |
| Minimum death 0-bars |  | ✓ | ✓ |  |  |
| Maximum death 0-bars | ✓ |  |  |  |  |
| Standard deviation death 0-bars | ✓ |  |  |  |  |
| Minimum birth 1-bars |  |  |  |  |  |
| Maximum birth 1-bars | ✓ |  |  |  |  |
| Minimum persistence 1-bars | ✓ | ✓ | ✓ | ✓ | ✓ |
| First quartile persistence 1-bars | ✓ |  |  |  |  |
| Maximum persistence 1-bars |  | ✓ |  |  |  |
| Mean birth/death 1-bars |  | ✓ | ✓ |  | ✓ |
| First quartile birth/death 1-bars |  | ✓ |  |  |  |
| Maximum birth/death 1-bars |  |  | ✓ |  |  |
| Total persistence 1-bars |  |  |  |  | ✓ |
| Number 0-bars | ✓ | ✓ | ✓ | ✓ | ✓ |
| Number 1-bars |  | ✓ | ✓ | ✓ |  |
| Entropy 0-bars |  | ✓ | ✓ |  |  |
| Total features | 8 | 9 | 8 | 4 | 5 |

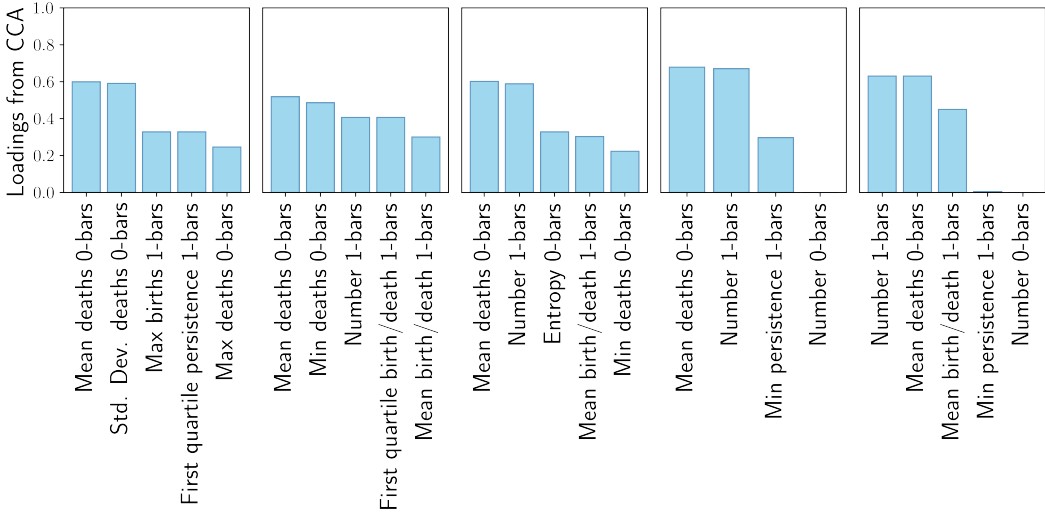

Figure 18: **CCA loadings for clean vs. poisoned activations**. Loadings of the 5 most important contributions to the first canonical variable of the CCA on the pruned barcode summaries show that the mean of the death of 0-bars is significantly correlated with the first two principal components of the PCA across all layers.

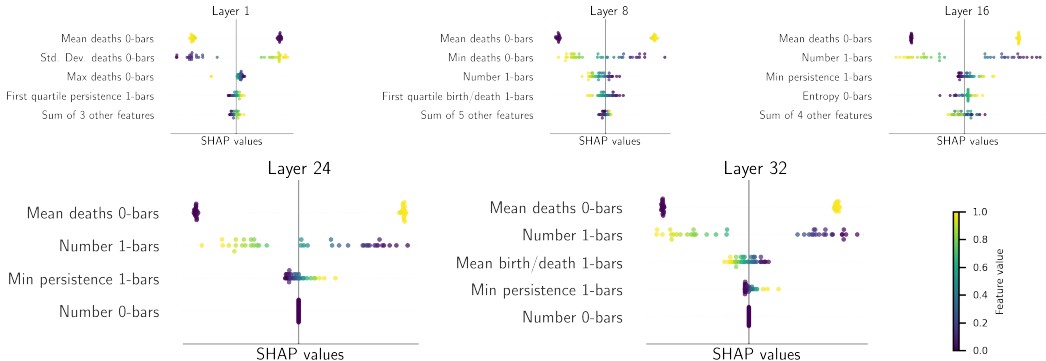

Figure 19: **SHAP analysis: clean vs. poisoned activations.** Beeswarm plot of logistic regression SHAP values trained on the pruned barcode summaries for layer 1, 8, 16, 24, and 32.

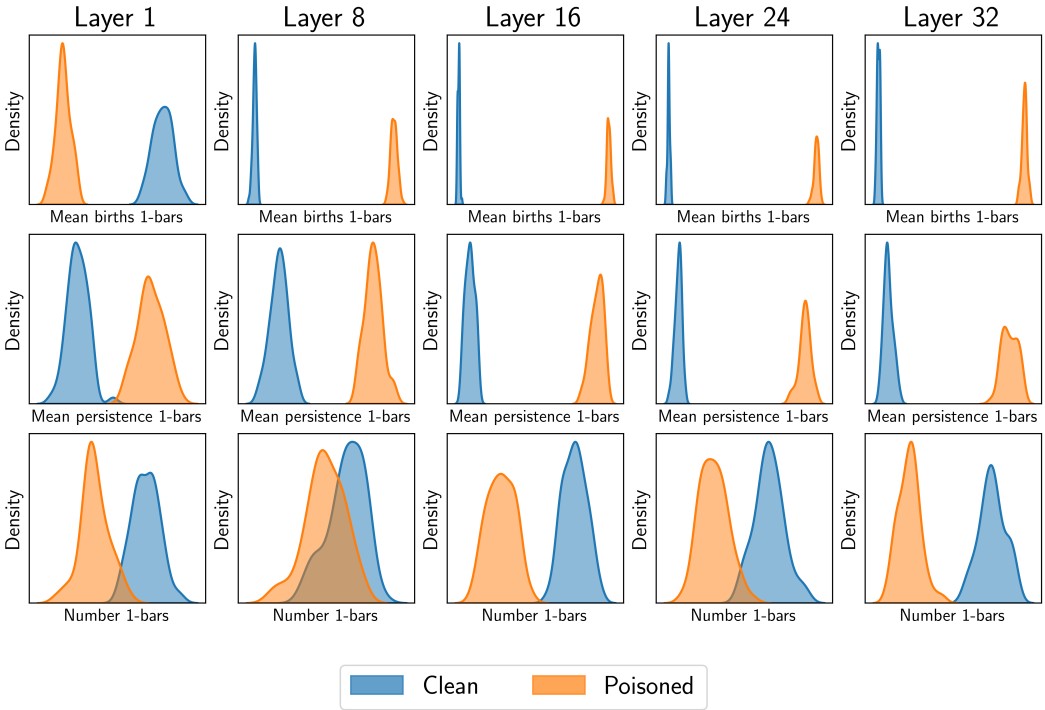

Figure 20: **Histograms for the mean of the births of 1-bars, mean persistence of 1-bars and number of 1-bars for Mistral.** Features extracted from the barcode summaries of the activations for layers 1, 8, 16, 24 and 32 of the clean vs. poisoned dataset.

### C.3.2    PHI3-MINI-4K (3.8B PARAMETERS)

We provide the results of the analysis depicted in Figure 3 including layers 1, 8, 16, 23, and 32 for Phi 3 (3.8B parameters) where barcodes are computed using the Euclidean distance in the representation space.

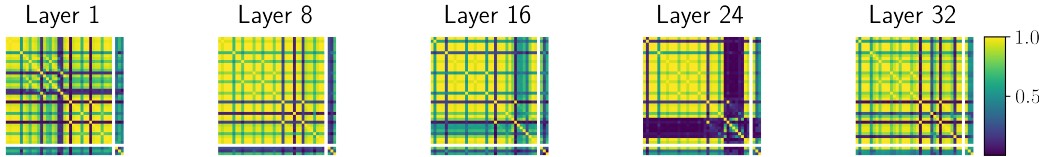

Figure 21: **Cross-correlation matrices for the barcode summaries for clean vs. poisoned activations.** Growing block of correlated features appears in the cross-correlation matrix of the barcode summaries appears in the middle layers (layers 1, 8, 16, 24, and 32 are shown).

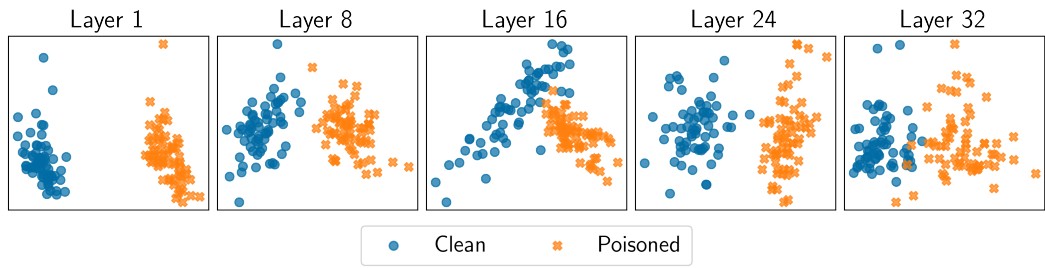

Figure 22: **PCA of barcode summaries of clean vs. poisoned activations**. Clear distinction appears in the projection onto the two first principal components from the PCA of the pruned barcode summaries for layers 1, 8, 16, 24, and 32.

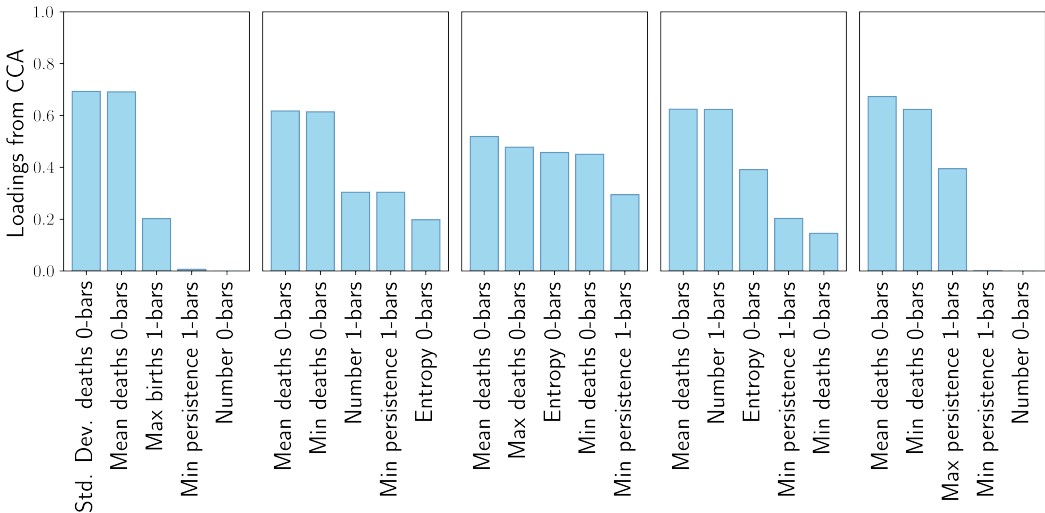

Figure 23: **CCA loadings for clean vs. poisoned activations**. Loadings of the 5 most important contributions to the first canonical variable of the CCA on the pruned barcode summaries show that the mean of the death of 0-bars is significantly correlated with the first two principal components of the PCA across all layers.

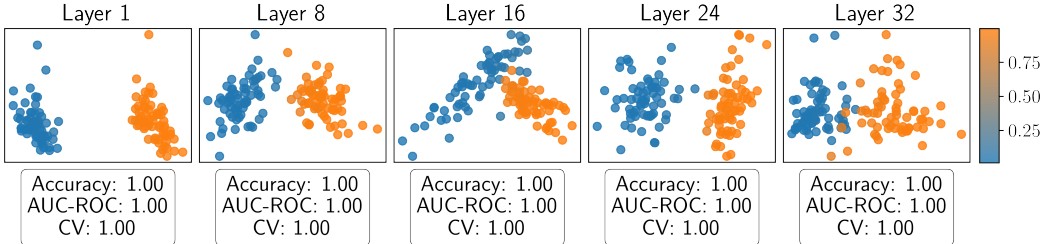

Figure 24: **Logistic regression for clean vs. poisoned activations.** Prediction of a logistic regression trained on a 70/30 train/test split of the pruned barcode summaries, plotted on the projection onto the two first principal components for visualization purposes. Accuracy and AUC–ROC tested on the test data, and 5-fold cross validation on train data are presented for each model, showcasing the outstanding performance of all models.

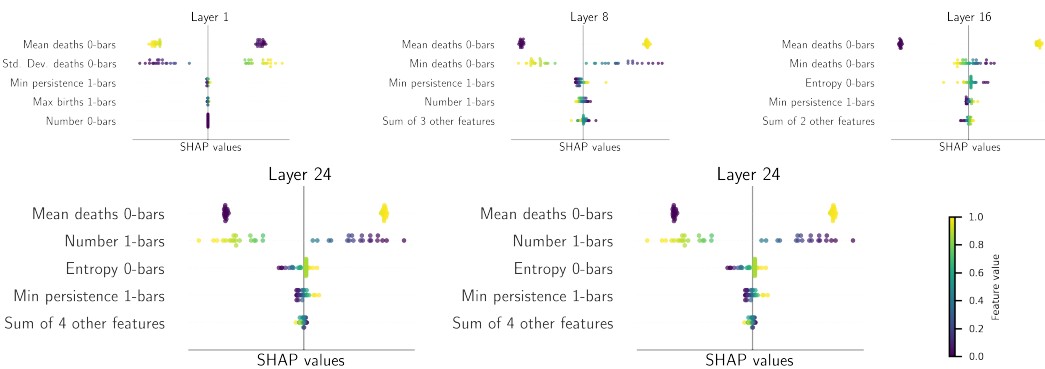

Figure 25: **SHAP analysis: clean vs. poisoned activations.** Beeswarm plot of logistic regression SHAP values trained on the pruned barcode summaries for layer 1, 8, 16, 24, and 32.

### C.3.3 MIXTRAL-8X7B (7B PARAMETERS)

We provide the results of the analysis depicted in Figure 3 including layers 1, 8, 16, 23 and 32 for the Mixtral 8 (7B parameters) model where barcodes are computed using the Euclidean distance in the representation space. We observe very similar results to the ones obtained with Mistral, indicating a consistency across models of the topological deformations of adversarial influence via XPIA (see Section 3.1).

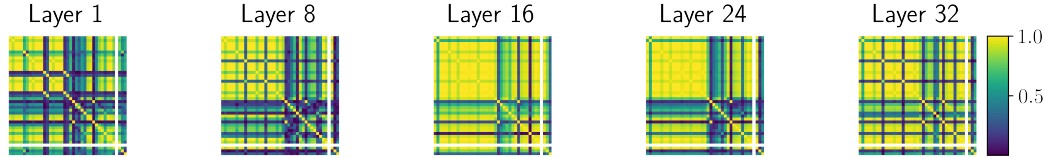

Figure 26: **Cross-correlation matrices for the barcode summaries for clean vs. poisoned activations.** Growing block of correlated features appears in the cross-correlation matrix of the barcode summaries for layers 1, 8, 16, 24, and 32. Correlations in layer 1 are lower than with Mistral 7B, see Figure 6.

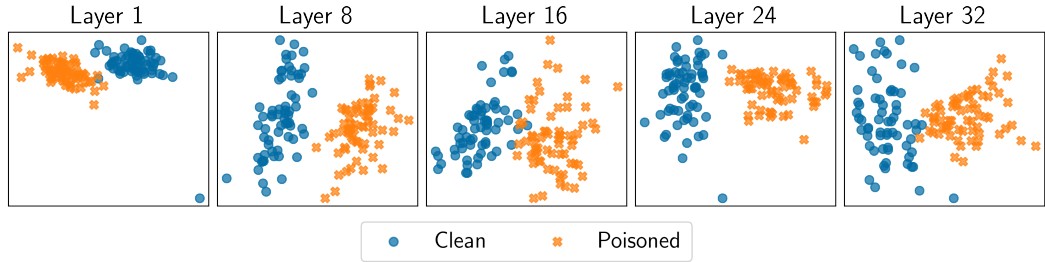

Figure 27: **PCA of barcode summaries of clean vs. poisoned activations**. Clear distinction appears in the projection onto the two first principal components from the PCA of the pruned barcode summaries for layers 1, 8, 16, 24, and 32.

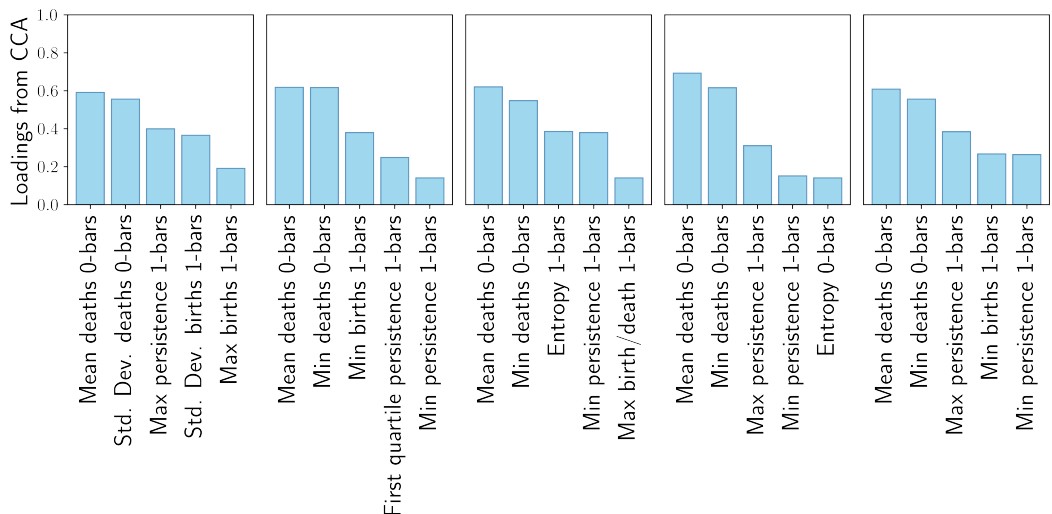

Figure 28: **CCA loadings for clean vs. poisoned activations**. Loadings of the 5 most important contributions to the first canonical variable of the CCA on the pruned barcode summaries show that the mean of the death of 0-bars is significantly correlated with the first two principal components of the PCA across all layers.

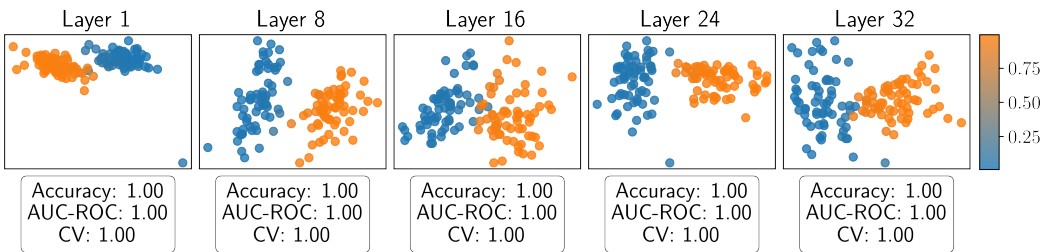

Figure 29: **Logistic regression for clean vs. poisoned activations.** Prediction of a logistic regression trained on a 70/30 train/test split of the pruned barcode summaries, plotted on the projection onto the two first principal components for visualization purposes. Accuracy and AUC–ROC tested on the test data, and 5-fold cross validation on train data are presented for each model, showcasing the outstanding performance of all models.

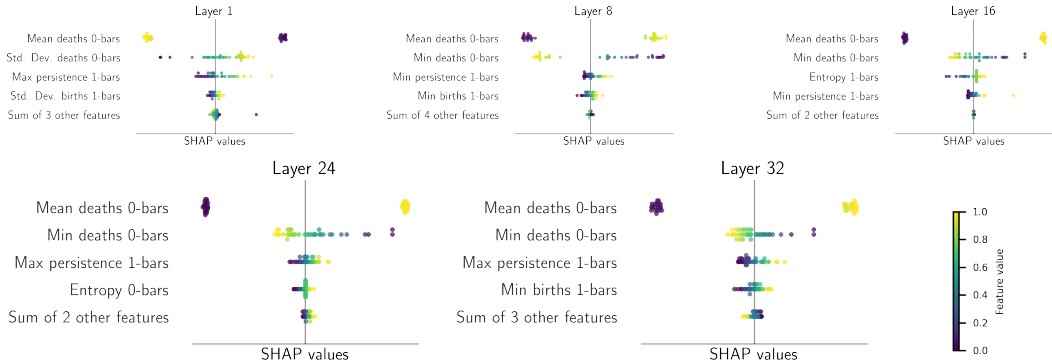

Figure 30: **SHAP analysis: clean vs. poisoned activations.** Beeswarm plot of logistic regression SHAP values trained on the pruned barcode summaries for layer 1, 8, 16, 24, and 32.

### C.3.4  LLaMA3 (8B PARAMETERS)

We provide the results of the analysis depicted in Figure 3 including layers 1, 8, 16, 23 and 32 for the Llama 3 (8B parameters) where barcodes are computed using the Euclidean distance in the representation space. We observe very similar results to the ones obtained with Mistral, indicating a consinstency across models of the topological deformations of adversarial influence via XPIA (see Section 3.1).

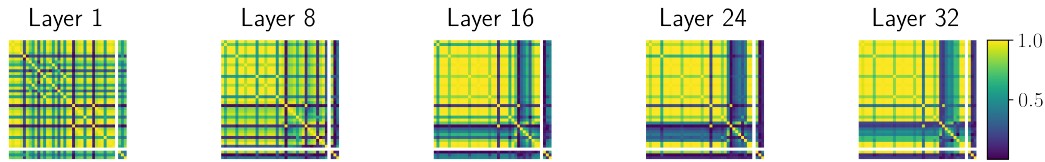

Figure 31: **Cross-correlation matrices for the barcode summaries for clean vs. poisoned activations.** Growing block of correlated features appears in the cross-correlation matrix of the barcode summaries for layers 1, 8, 16, 24, and 32. Correlations in layer 1 are lower than with Mistral 7B, see Figure 6.

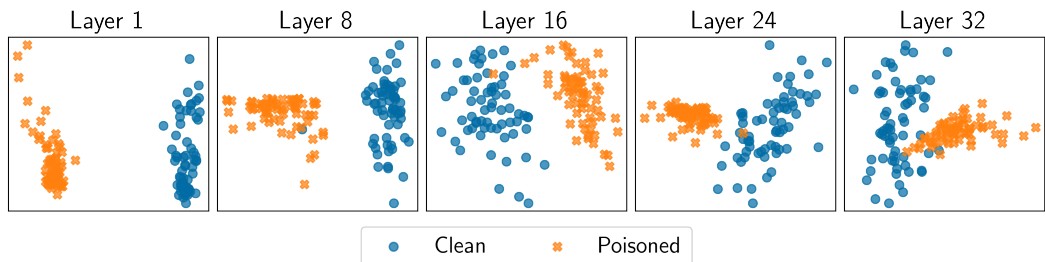

Figure 32: **PCA of barcode summaries of clean vs. poisoned activations**. Clear distinction appears in the projection onto the two first principal components from the PCA of the pruned barcode summaries for layers 1, 8, 16, 24, and 32.

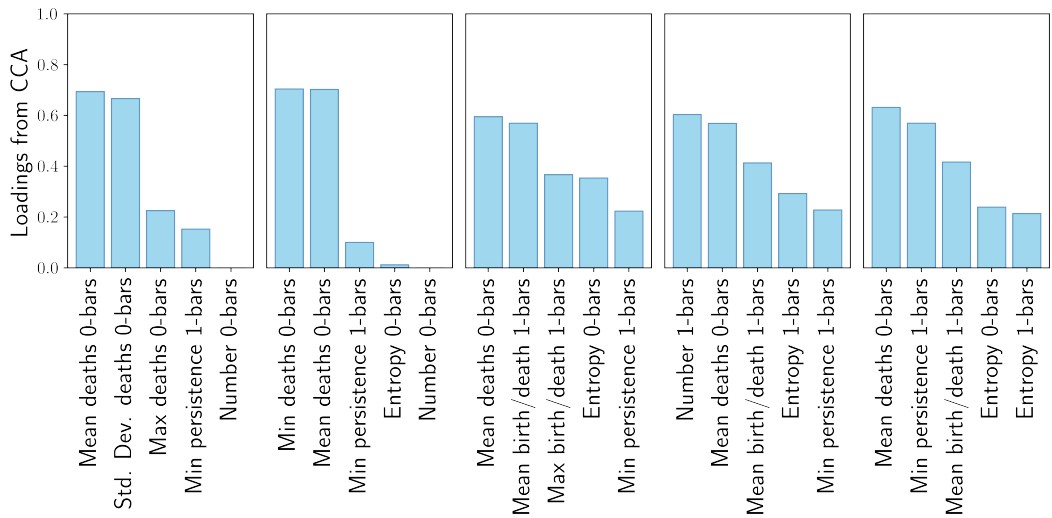

Figure 33: **CCA loadings for clean vs. poisoned activations**. Loadings of the 5 most important contributions to the first canonical variable of the CCA on the pruned barcode summaries show that the mean of the death of 0-bars is significantly correlated with the first two principal components of the PCA across all layers.

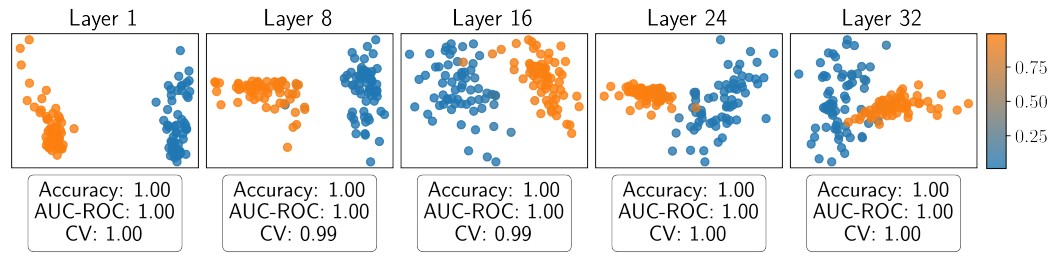

Figure 34: **Logistic regression for clean vs. poisoned activations.** Prediction of a logistic regression trained on a 70/30 train/test split of the pruned barcode summaries, plotted on the projection onto the two first principal components for visualization purposes. Accuracy and AUC–ROC tested on the test data, and 5-fold cross validation on train data are presented for each model, showcasing the outstanding performance of all models.

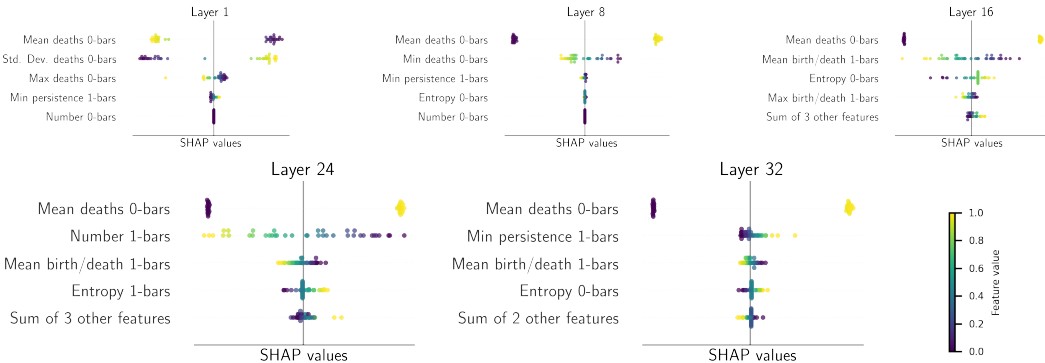

Figure 35: **SHAP analysis: clean vs. poisoned activations.** Beeswarm plot of logistic regression SHAP values trained on the pruned barcode summaries for layer 1, 8, 16, 24, and 32.

### C.3.5 PHI3-MEDIUM-128K (14B PARAMETERS)

We provide the results of the analysis depicted in Figure 3 including layers 1, 8, 16, 23 and 32 for the Phi-3-medium (14B parameters) model where barcodes are computed using the Euclidean distance in the representation space. We observe very similar results to the ones obtained with Mistral, indicating a consistency across models of the topological deformations of adversarial influence via XPIA (see Section 3.1).

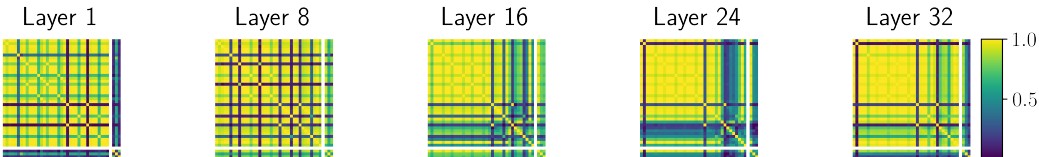

Figure 36: **Cross-correlation matrices for the barcode summaries for clean vs. poisoned activations.** Growing block of correlated features appears in the cross-correlation matrix of the barcode summaries for layers 1, 8, 16, 24, and 32. Correlations in layer 1 are lower than with Mistral 7B, see Figure 6.

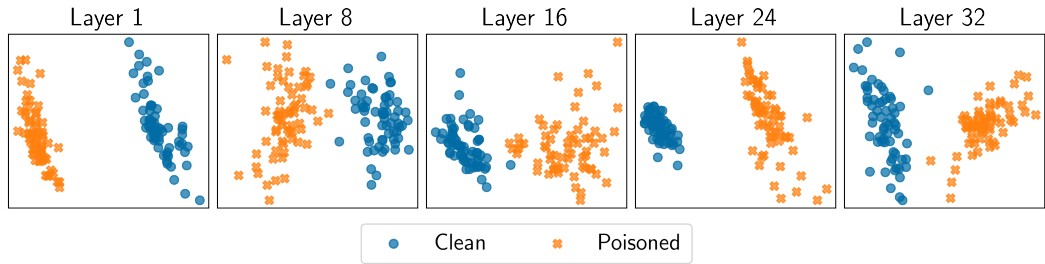

Figure 37: **PCA of barcode summaries of clean vs. poisoned activations**. Clear distinction appears in the projection onto the two first principal components from the PCA of the pruned barcode summaries for layers 1, 8, 16, 24, and 32.

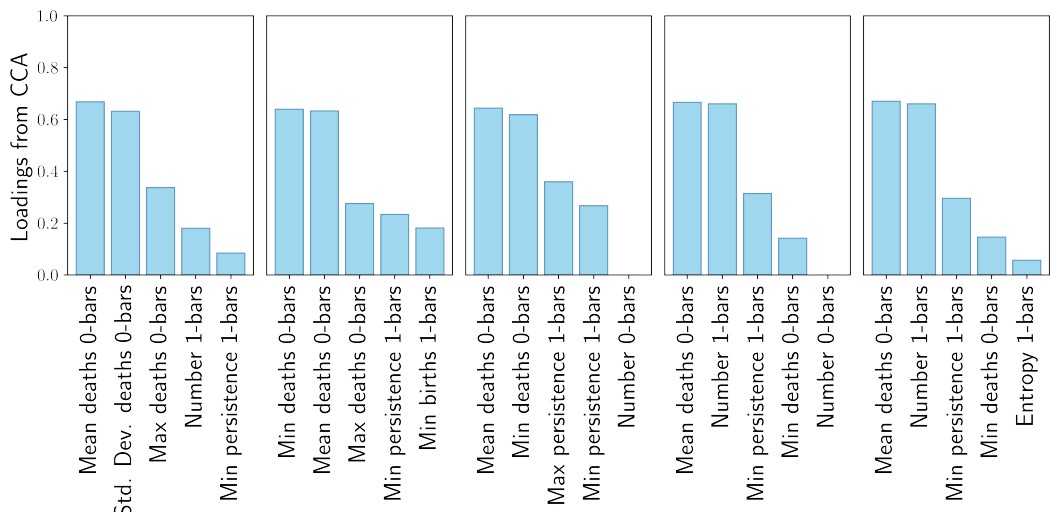

Figure 38: **CCA loadings for clean vs. poisoned activations**. Loadings of the 5 most important contributions to the first canonical variable of the CCA on the pruned barcode summaries show that the mean of the death of 0-bars is significantly correlated with the first two principal components of the PCA across all layers.

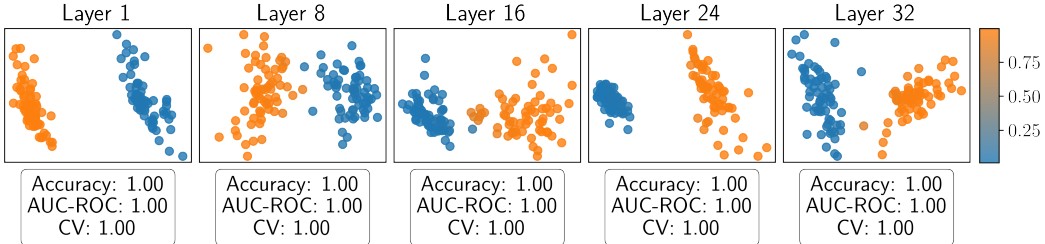

Figure 39: **Logistic regression for clean vs. poisoned activations.** Prediction of a logistic regression trained on a 70/30 train/test split of the pruned barcode summaries, plotted on the projection onto the two first principal components for visualization purposes. Accuracy and AUC–ROC tested on the test data, and 5-fold cross validation on train data are presented for each model, showcasing the outstanding performance of all models.

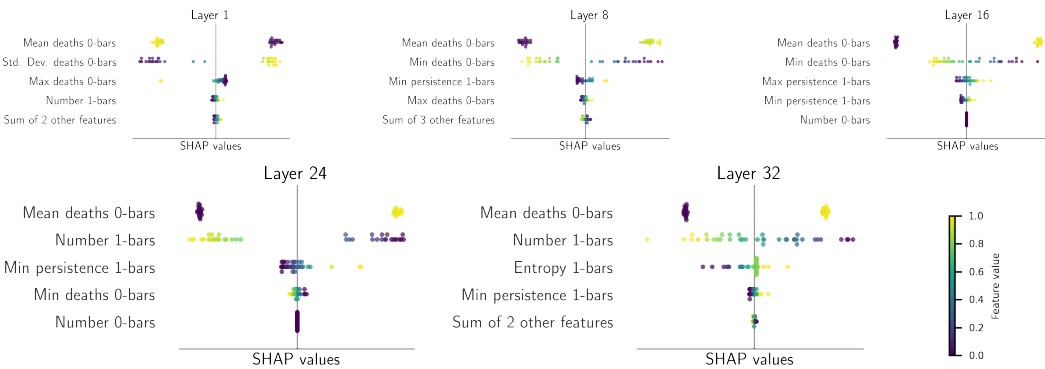

Figure 40: **SHAP analysis: clean vs. poisoned activations.** Beeswarm plot of logistic regression SHAP values trained on the pruned barcode summaries for layer 1, 8, 16, 24, and 32.

#### C.3.6    LLAMA3 (70B PARAMETERS)

We provide the results of the analysis depicted in Figure 3 including layers 1, 8, 16, 23 and 32 for the Llama 3 (70B parameters) where barcodes are computed using the Euclidean distance in the representation space. We observe very similar results to the ones obtained with Mistral, indicating a consinstency across models of the topological deformations of adversarial influence via XPIA (see Section 3.1).

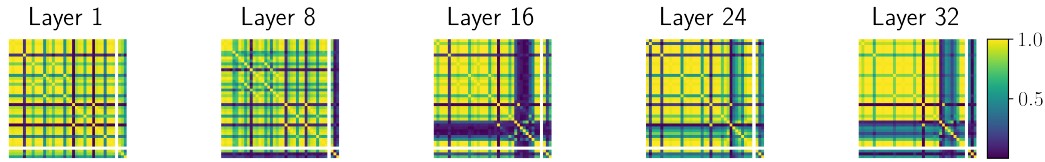

Figure 41: **Cross-correlation matrices for the barcode summaries for clean vs. poisoned activations.** Growing block of correlated features appears in the cross-correlation matrix of the barcode summaries for layers 1, 8, 16, 24, and 32. Correlations in layer 1 are lower than with Mistral 7B, see Figure 6.

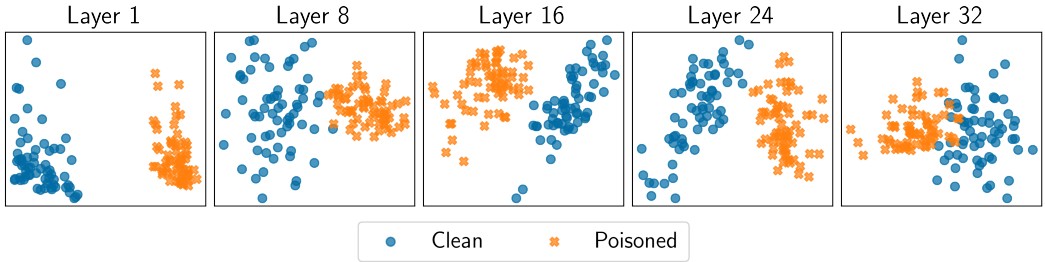

Figure 42: **PCA of barcode summaries of clean vs. poisoned activations**. Clear distinction appears in the projection onto the two first principal components from the PCA of the pruned barcode summaries for layers 1, 8, 16, 24, and 32.

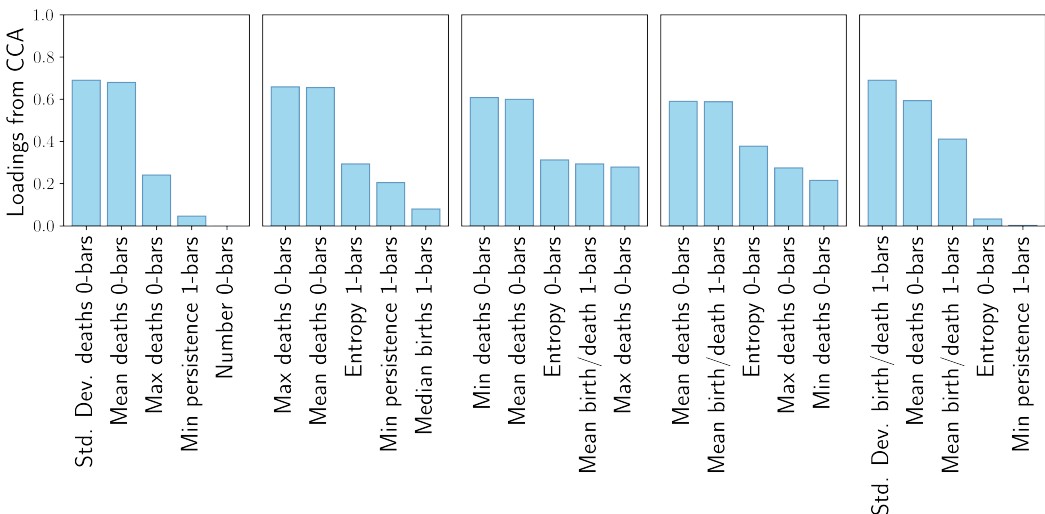

Figure 43: **CCA loadings for clean vs. poisoned activations**. Loadings of the 5 most important contributions to the first canonical variable of the CCA on the pruned barcode summaries show that the mean of the death of 0-bars is significantly correlated with the first two principal components of the PCA across all layers.

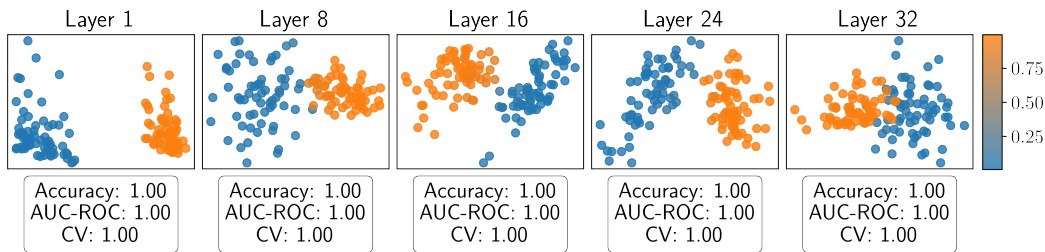

Figure 44: **Logistic regression for clean vs. poisoned activations.** Prediction of a logistic regression trained on a 70/30 train/test split of the pruned barcode summaries, plotted on the projection onto the two first principal components for visualization purposes. Accuracy and AUC–ROC tested on the test data, and 5-fold cross validation on train data are presented for each model, showcasing the outstanding performance of all models.

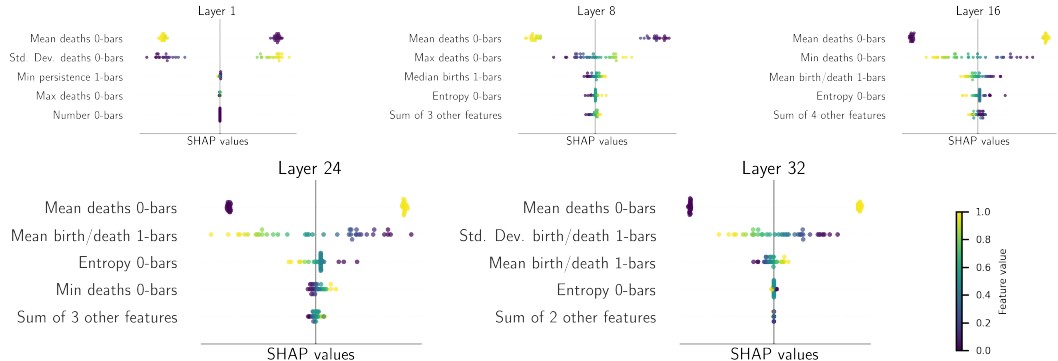

Figure 45: **SHAP analysis: clean vs. poisoned activations.** Beeswarm plot of logistic regression SHAP values trained on the pruned barcode summaries for layer 1, 8, 16, 24, and 32.

## C.4 Results: Locked vs. Elicited

### C.4.1 Mistral 7B

We include the results of the global analysis in Figure 3 for the locked vs. elicited dataset. There are two main differences with previous results: the block of high correlated features presents a less clear trend and is more faint in layer 16, resulting in the need of more features in the analysis; and the mean death of the 0-bars changes the sign of its influence in classifying locked and elicited models across layers. However the distinction in the PCA of the barcode summaries remains clear and the logistic regression still achieves perfect performance, despite a slightly less straightforward analysis.

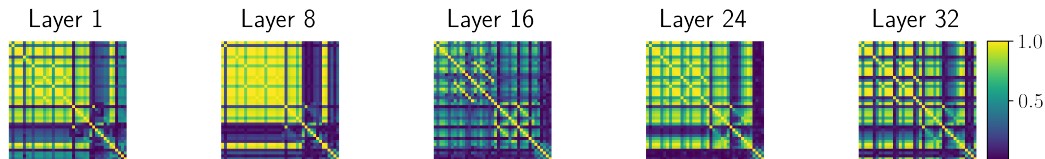

Figure 46: **Mistral with Euclidean distance: Cross-correlation matrices for the barcode summaries for locked vs. elicited activations.** Growing block of correlated features appears in the cross-correlation matrix of the barcode summaries for layers 1, 8, 16, 24, and 32.

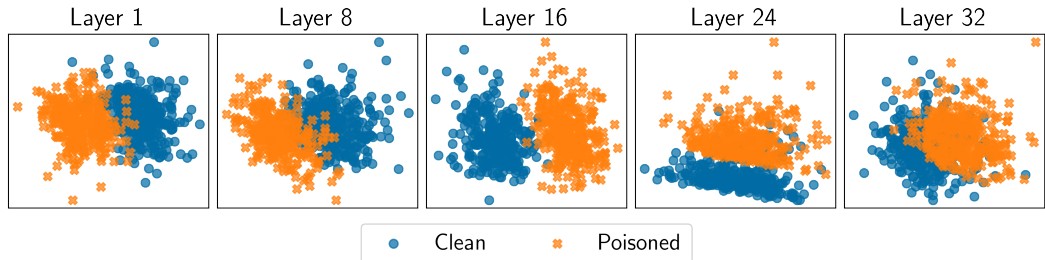

Figure 47: **Mistral with Euclidean distance: PCA of barcode summaries of locked vs. elicited activations**. Clear distinction appears in the projection onto the two first principal components from the PCA of the pruned barcode summaries for layers 1, 8, 16, 24, and 32.

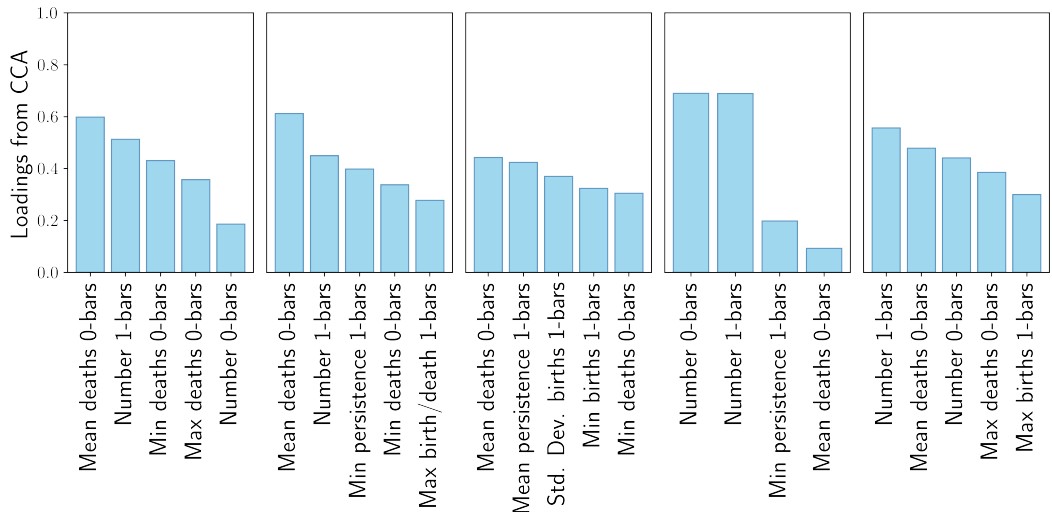

Figure 48: **Mistral with Euclidean distance: CCA loadings for locked vs. elicited activations**. Loadings of the 5 most important contributions to the first canonical variable of the CCA on the pruned barcode summaries show that the mean of the death of 0-bars is significantly correlated with the first two principal components of the PCA across all layers.

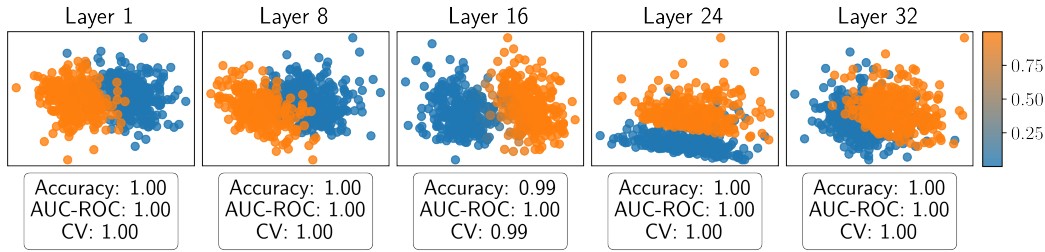

Figure 49: **Mistral with Euclidean distance: Logistic regression for locked vs. elicited activations.** Prediction of a logistic regression trained on a 70/30 train/test split of the pruned barcode summaries, plotted on the projection onto the two first principal components for visualization purposes. Accuracy and AUC–ROC tested on the test data, and 5-fold cross validation on train data are presented for each model, showcasing the outstanding performance of all models.

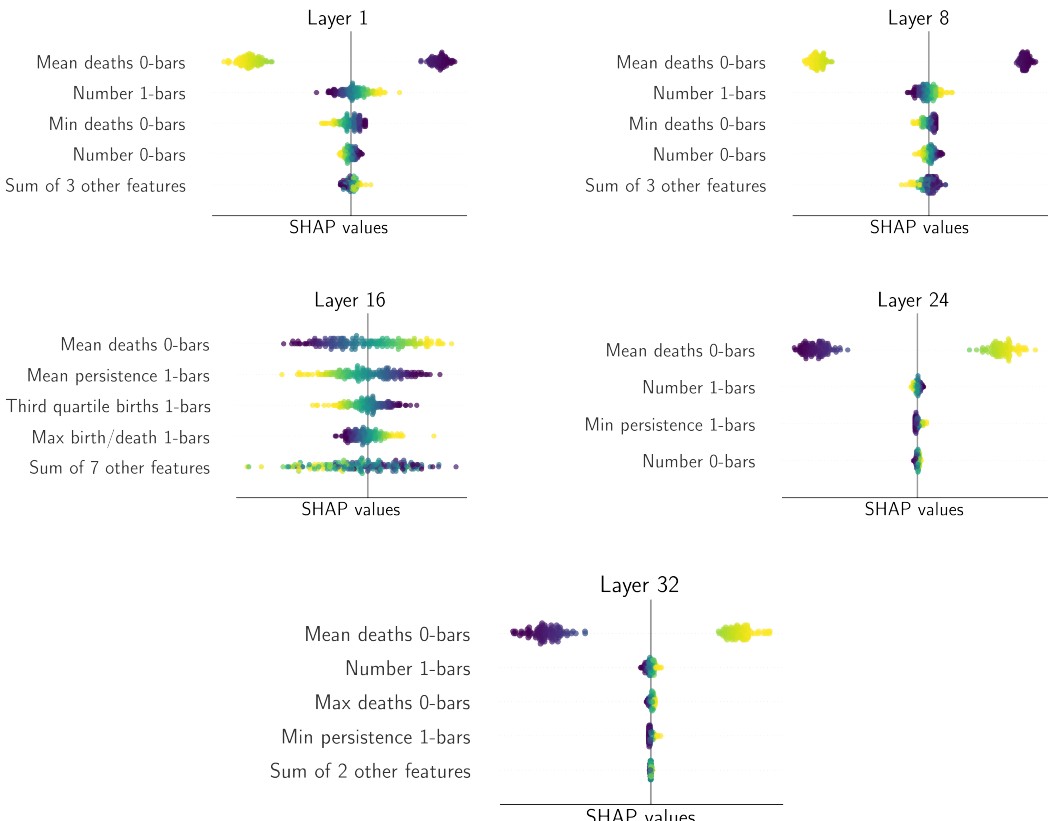

Figure 50: **Mistral with Euclidean distance: SHAP analysis for locked vs. elicited activations.** Beeswarm plot of the SHAP values for the logistic regression trained on the pruned barcode summaries for layer 1, 8, 16, 24, and 32. The mean of the deaths of 0-bars appears as the most impactful feature in the prediction of the model, shifting predictions to "locked" when the value of the feature is lower for layers 8, 16, 23, and 32, and to "elicited" when it is higher. The opposite phenomenon is observed in layer 0.

### C.4.2 LLAMA3 (8B PARAMETERS)

We include the results of the global analysis in Figure 3 for the locked vs. elicited dataset. Here we also observe less clear patterns of correlations in the topological features, particularly for latter layers. Despite the mean of the death of 0-bars remaining as one of the key features in the CCA, the interpretation of the Shapley values is less straightforward in this case as the dichotomous behavior of these for the mean of the 0-bars disappears for latter layers.

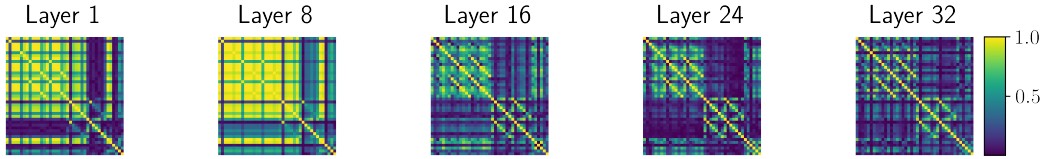

Figure 51: **Llama with Euclidean distance: Cross-correlation matrices for the barcode summaries for locked vs. elicited activations.** Decreasing block of correlated features appears in the cross-correlation matrix of the barcode summaries for layers 1, 8, 16, 24, and 32.

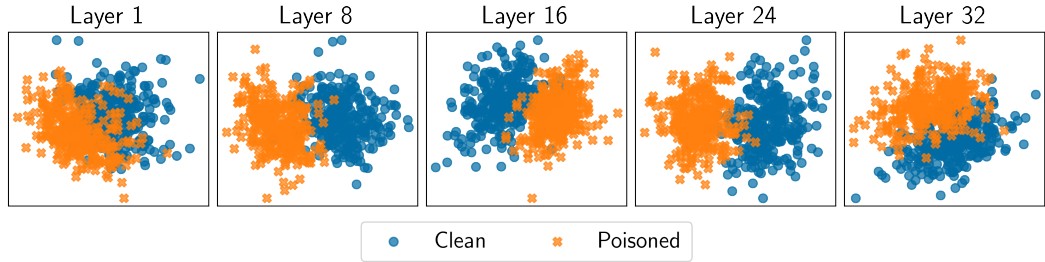

Figure 52: **Llama with Euclidean distance: PCA of barcode summaries of locked vs. elicited activations**. Clear distinction appears in the projection onto the two first principal components from the PCA of the pruned barcode summaries for layers 1, 8, 16, 24, and 32.

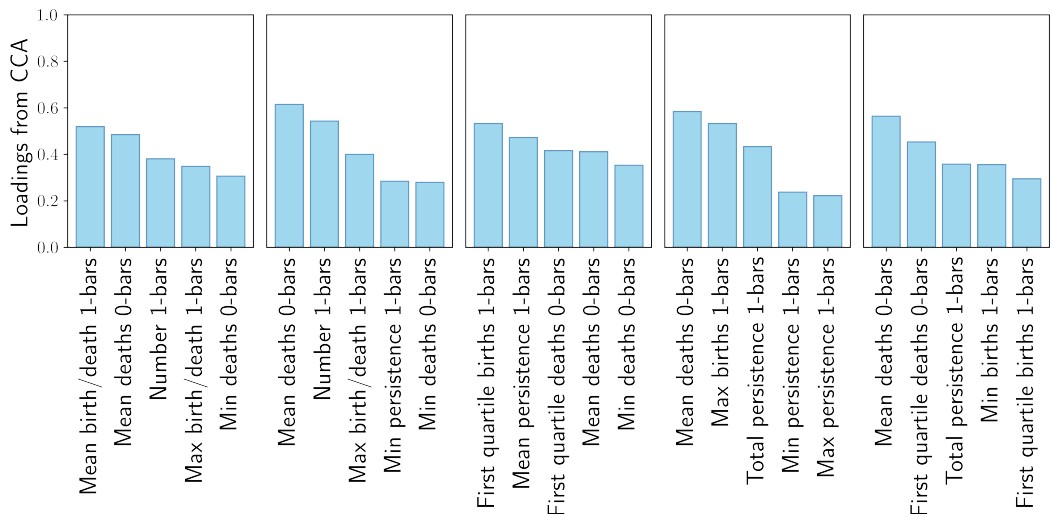

Figure 53: **Llama with Euclidean distance: CCA loadings for locked vs. elicited activations**. Loadings of the 5 most important contributions to the first canonical variable of the CCA on the pruned barcode summaries show that the mean of the death of 0-bars is significantly correlated with the first two principal components of the PCA across all layers.

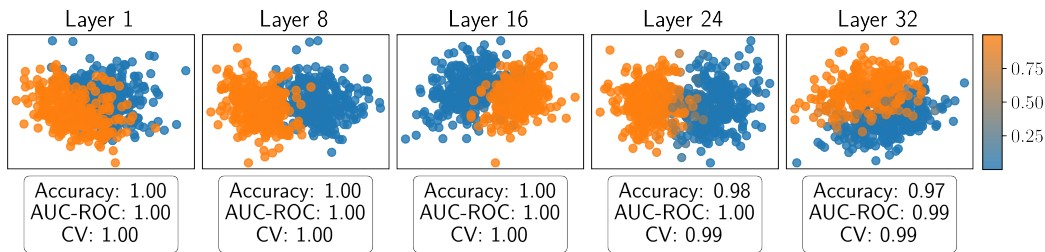

Figure 54: **Llama with Euclidean distance: Logistic regression for locked vs. elicited activations.** Prediction of a logistic regression trained on a 70/30 train/test split of the pruned barcode summaries, plotted on the projection onto the two first principal components for visualization purposes. Accuracy and AUC–ROC tested on the test data, and 5-fold cross validation on train data are presented for each model, showcasing the outstanding performance of all models.

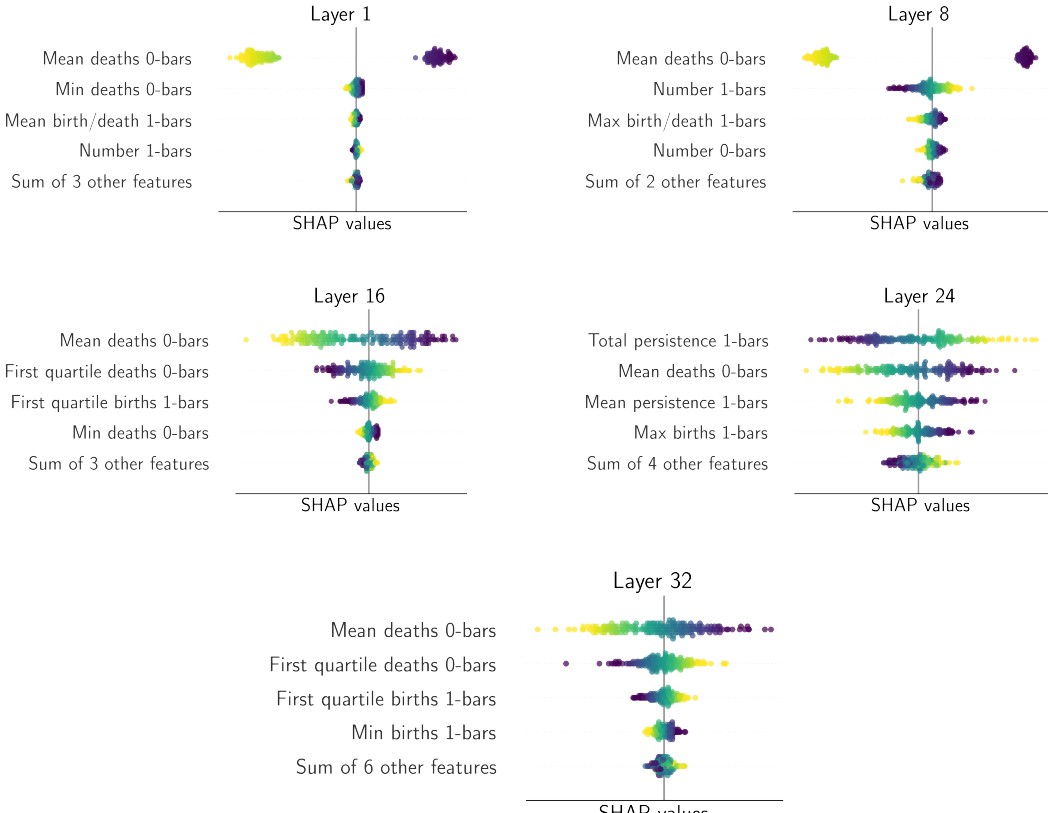

Figure 55: **Mistral with Euclidean distance: SHAP analysis for locked vs. elicited activations.** Beeswarm plot of the SHAP values for the logistic regression trained on the pruned barcode summaries for layer 1, 8, 16, 24, and 32. The mean of the deaths of 0-bars appears as the most impactful feature in the prediction of the model, shifting predictions to "locked" when the value of the feature is lower for layers 8, 16 and 32, and to "elicited" when it is higher. For layer 24, the total persistence of 1-bars appears as the most important feature. Lower number of 1-bars classifies the point as "locked" while higher values push the prediction toward "elicited".

## D  FURTHER DETAILS ON LOCAL ANALYSIS

In this section we provide further details to the local analysis in Section 3.3.

### D.1  PIPELINE

Within this local analysis, we aim to determine the interaction of elements of the neural network across the layers by taking representations across pairs of layers as coordinates in 2 dimensions (2D). We study this across three models: Mistral, Phi3 3.8B and LLaMA3 8B. For each of these models, we take a sample of 2000 from each model, 1000 of which are clean activations and 1000 of which are poisoned activations. Each element along the layer given their embedding into 2D can be thought of as nodes in a graph with weighted connections based on the Euclidean distances between the points. On these graphs, we construct the Vietoris–Rips filtration and compute the resulting persistence barcode which describes the topology of the interactions between the elements.

For this local analysis, we focus on a smaller selection of persistence barcode summaries, including measures such as the mean death of 0-bars, total persistence of 0- and 1-bars, and persistent entropy, while excluding measures such as the quantiles of death bars. We compute these summary statistics and track their progression across pairs of layers in the models. We presented one such progression within Figure 10 in Section 3.3, which captures how total persistence changes over the layers and

is distinct from the control case. In the following sections, we include further plots to support this argument.

## D.2 RESULTS

### D.2.1 MISTRAL MODEL

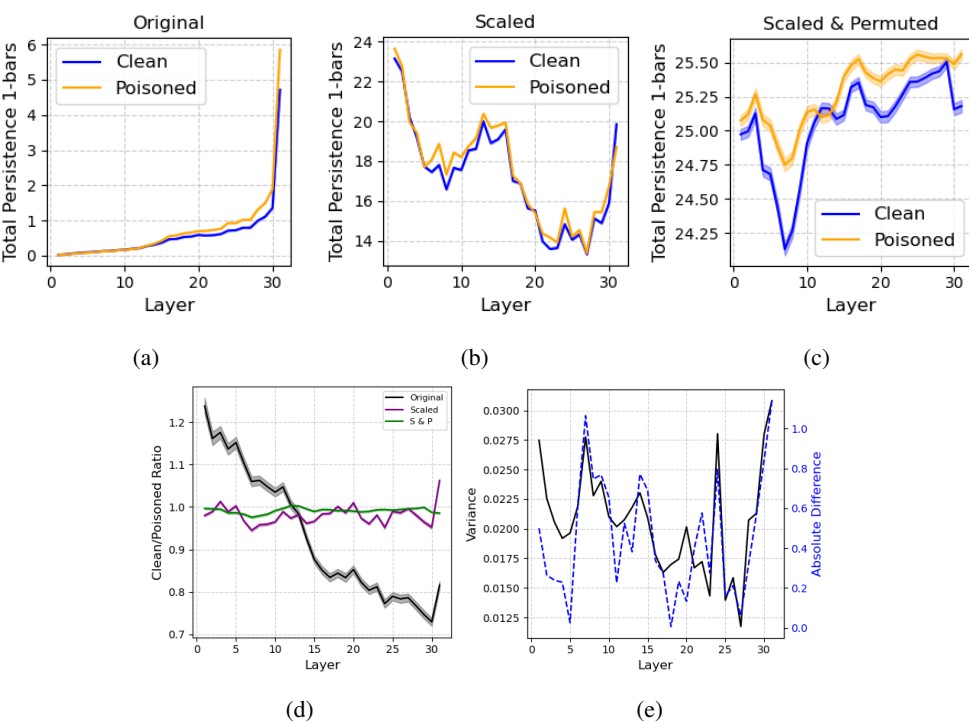

(a)  (b)  (c)

(d)  (e)

Figure 56: **Local analysis of consecutive layers for the total persistence of 1-bars for the Mistral model.** Comparisons of the average total persistence of 1-bars across 1000 samples for Mistral model for original **(a)**, scaled/normalized **(b)** and scaled and permuted **(c)** activation data. **(d)** Ratios of mean total persistence of 1-bars between clean and poisoned datasets for original, scaled, and scaled and permuted activations. **(e)** Overlaid plots of the overall variance of total persistence of 1-bars for clean and poisoned datasets combined and the absolute difference between mean total persistence of 1-bars for clean and poisoned datasets.

In addition to the propagation of total persistence of 1-bars we showed in Section 3.3 and in this section of the Appendix, we also evaluated the progression of other barcode summaries. Notably, descriptors which capture similar features are the mean deaths of 1-bars, and the mean birth of 0 bars with mirroring patterns. In Figure 57, we show the results for the mean death of 0-bars.

### D.2.2 PHI3 MODEL

We present a similar comparison of results for the Phi3 model. Figure 58 illustrates the patterns across layers for the mean death of 0-bars, while Figure 59 shows the patterns for the total persistence of 1-bars. Unlike the Mistral model, the ratio between barcode statistics for clean and poisoned activations in the Phi3 model does not intersect one. While a decreasing or somewhat parabolic trend is still observed, the average mean death of 0-bars and the total persistence of 1-bars for clean raw activations consistently remain greater than those for poisoned raw activations. Additionally, we find that the "control" case remains close to the x-axis, with the scaled ratios exhibiting significant variations around this baseline.

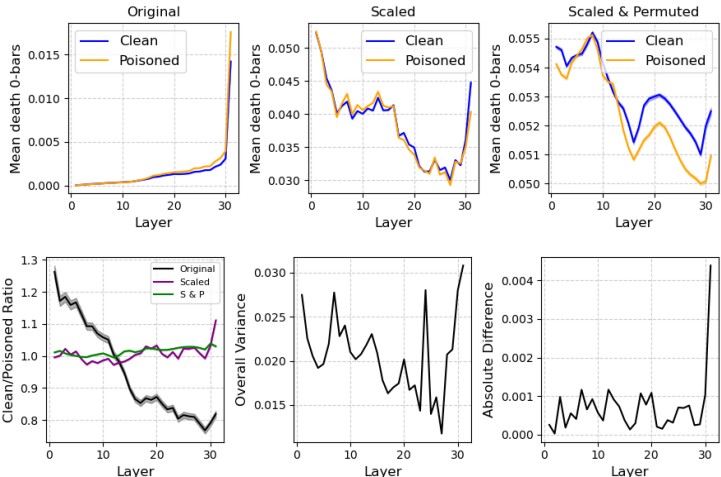

Figure 57: **Local analysis of consecutive layers for the mean deaths of 0-bars for the Mistral model. Top:** Comparisons of the average of mean deaths of 0-bars across 1000 samples for the Mistral model for original (raw), scaled (normalized) and scaled & permuted activation data. **Bottom left:** Ratios of average mean deaths of 0-bars between clean and poisoned datasets for original, scaled and scaled & permuted activations. **Bottom center:** Overall variance of mean deaths of 0-bars for clean and poisoned datasets combined. **Bottom right:** Absolute difference between mean total persistence of 1-bars for clean and poisoned datasets.

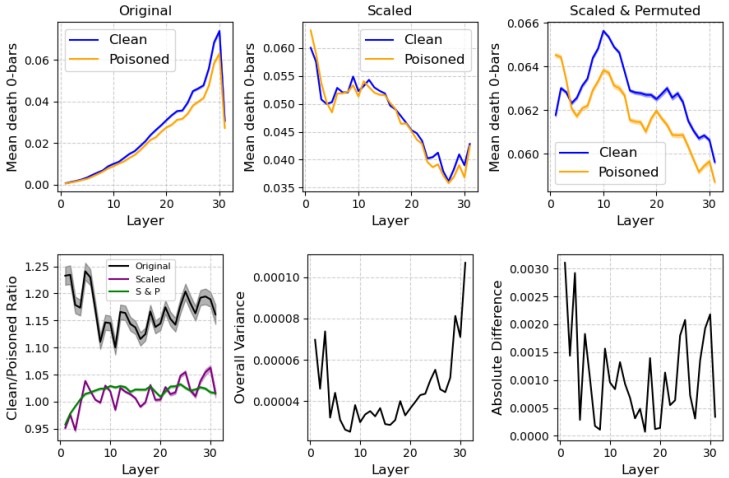

Figure 58: **Local analysis of consecutive layers for the mean deaths of 0-bars for the Phi3 model. Top:** Comparisons of the average of mean deaths of 0-bars across 1000 samples for Phi3 model for original (raw), scaled (normalized) and scaled & permuted activation data. **Bottom left:** Ratios of average mean deaths of 0-bars between clean and poisoned datasets for original, scaled and scaled & permuted activations. **Bottom center:** Overall variance of mean deaths of 0-bars for clean and poisoned datasets combined. **Bottom right:** Absolute difference between mean total persistence of 1-bars for clean and poisoned datasets.

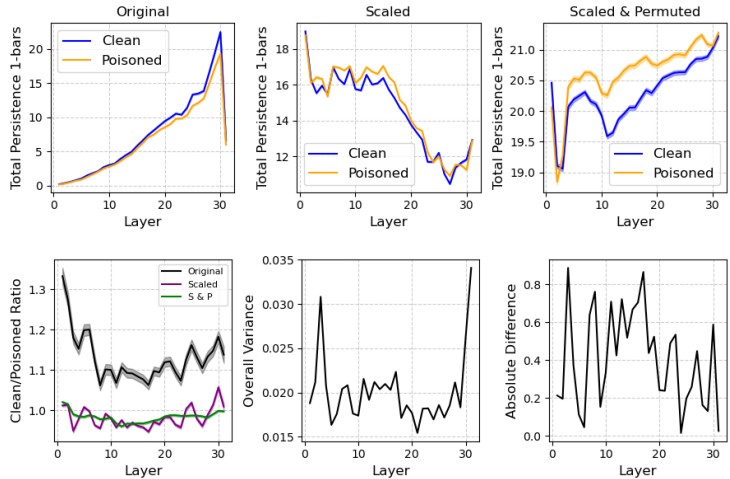

Figure 59: **Local analysis of consecutive layers for the total persistence of 1-bars for the Phi3 model. Top:** Comparisons of the average of total persistence of 1-bars across 1000 samples for Phi3 model for original (raw), scaled (normalized) and scaled & permuted activation data. **Bottom left:** Ratios of average total persistence of 1-bars between clean and poisoned datasets for original, scaled and scaled & permuted activations. **Bottom center:** Overall variance of total persistence of 1-bars for clean and poisoned datasets combined. **Bottom right:** Absolute difference between mean total persistence of 1-bars for clean and poisoned datasets.

### D.2.3 LLaMA3 8B Model

We present the results for the LLaMA3 8B model. Figures 60 and 61 both show a decreasing trend in the ratio between clean and poisoned activations, whether measured by the mean death of 0-bars or the total persistence of 1-bars respectively. Notably, this ratio crosses 1 around layer 15 or later. Moreover, we continue to observe distinct differences between clean and poisoned activations across both meaningful variants.

### D.2.4 Peak Analysis for Phi3 and LLaMA3

Table 7: **Peak analysis.** Precision@$k$ for $k$=1, 3, and 5 largest peaks in total variance, and their precision in detecting the largest peaks in absolute difference between the two classes. Spearman's rank correlation ($r$) is reported in the last column. *, ** correspond to $p$-values $<.05$ and $.01$, respectively.

| Phi3 | p@1 | p@3 | p@5 | r |
|---|---|---|---|---|
| Total Persistence 0-bars | 0 | .33 | .2 | 0.69** |
| Total Persistence 1-bars | 1.0 | .67* | .8** | 0.50** |
| Mean Birth 1-bars | 0 | .33 | .6* | 0.66** |
| Mean Death 1-bars | 0 | .67* | .8** | 0.35 |
| LLAMA3 | p@1 | p@3 | p@5 | r |
| Total Persistence 0-bars | 1.0* | .33 | .4 | 0.60** |
| Total Persistence 1-bars | 1.0* | .67 | .8** | 0.93** |
| Mean Birth 1-bars | 1.0* | .67 | .6 | 0.60** |
| Mean Death 1-bars | 1.0* | .67* | .8* | 0.93** |

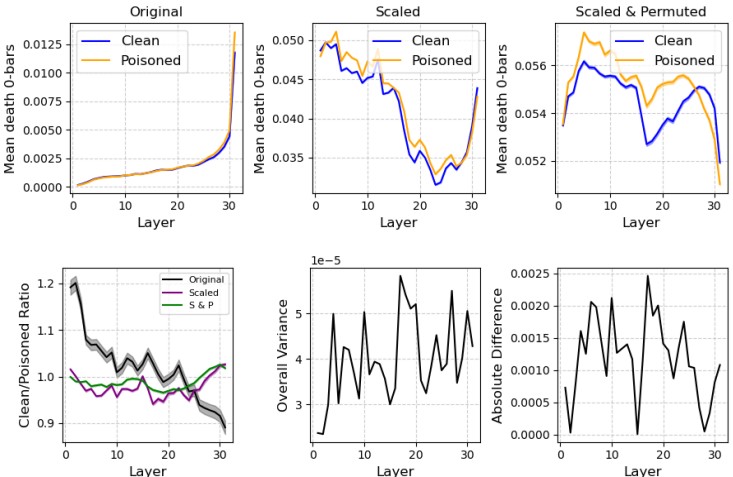

Figure 60: **Local analysis of consecutive layers for the mean deaths of 0-bars for the LLaMA3 8B model. Top:** Comparisons of the average of mean deaths of 0-bars across 1000 samples for LLaMA3 8B model for original (raw), scaled (normalized) and scaled & permuted activation data. **Bottom left:** Ratios of average mean deaths of 0-bars between clean and poisoned datasets for original, scaled and scaled & permuted activations. **Bottom center:** Overall variance of mean deaths of 0-bars for clean and poisoned datasets combined. **Bottom right:** Absolute difference between mean total persistence of 1-bars for clean and poisoned datasets.

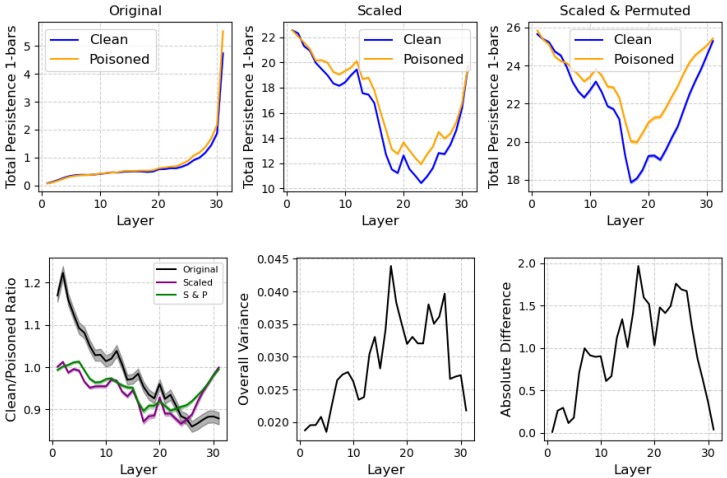

Figure 61: **Local analysis of consecutive layers for the total persistence of 1-bars for the LLaMA3 8B model. Top:** Comparisons of the average of total persistence of 1-bars across 1000 samples for the LLaMA3 8B model for original (raw), scaled (normalized) and scaled & permuted activation data. **Bottom left:** Ratios of average total persistence of 1-bars between clean and poisoned datasets for original, scaled and scaled & permuted activations. **Bottom center:** Overall variance of total persistence of 1-bars for clean and poisoned datasets combined. **Bottom right:** Absolute difference between mean total persistence of 1-bars for clean and poisoned datasets.

### D.2.5 NON-CONSECUTIVE LAYER ANALYSIS

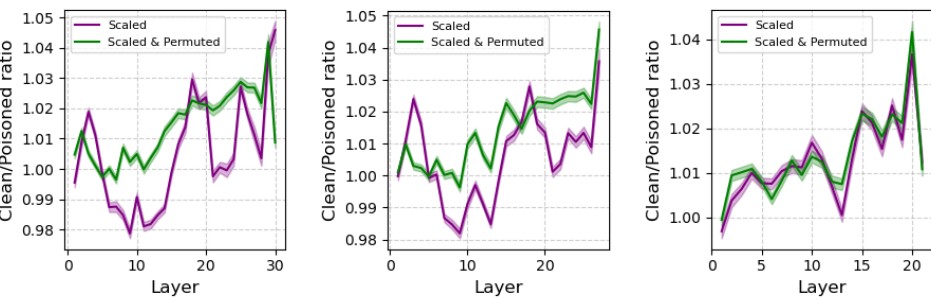

Figure 62: **Local analysis of non-consecutive layers for mean death of 0-bars.** Comparison of ratios between mean death of 0-bars for clean and poisoned datasets when considering topology pairs of layers at 1 (left), 3 (middle), and 10 (right) intervals apart.

Continuing the analysis of non-consecutive layers, we examine how increasing layer separation affects the contrast between clean and poisoned activations across different barcode summaries. Figure 62 shows the ratio of the mean death times of 0-bars, while Figure 63 shows the ratio of the total persistence of 1-bars. For both summaries, we observe that at a 10-layer separation, the similarity between the ratios for the scaled and control settings becomes more pronounced compared to separations of 1 or 3 layers, indicating a reduced distinction between clean and poisoned interactions at larger layer intervals.

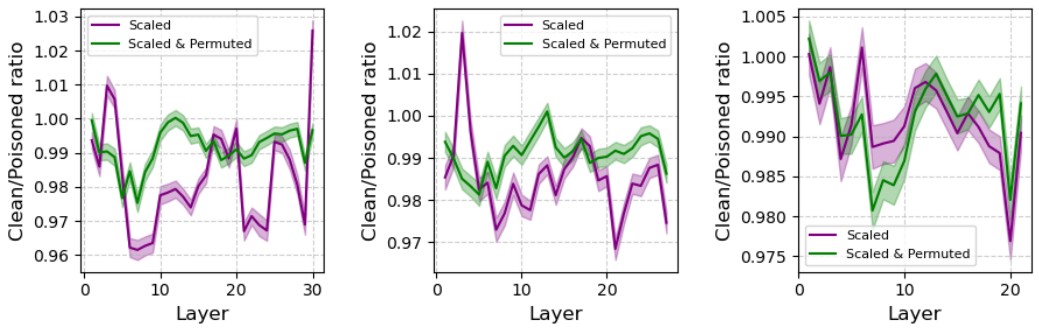

Figure 63: **Local analysis of non-consecutive layers on total persistence of 1-bars for the Mistral model.** Comparison of the ratios between total persistence of 1-bars for clean and poisoned datasets when considering the topology of interactions between pairs of layers at 1 (left), 3 (middle), and 10 (right) intervals apart.

## E SANDBAGGED REPRESENTATION DATA COLLECTION

To study sandbagging, we follow the "password-locking" paradigm introduced in prior work (Greenblatt et al., 2024; van der Weij et al., 2024), wherein a model is fine-tuned to hide certain behaviors or capabilities unless presented with a predefined password in the prompt. We leverage the the synthetic data generation process and fine-tuning procedure detailed in Section 3 and 4 of van der Weij et al. (2024). For clarity of our experimental setup, we summarize it briefly here. We extend it to extract final token hidden representations from the model under both elicited and elicited conditions.

1. We generate a synthetic dataset of 3,347 evaluation questions based on the the Weapons of Mass Destruction Proxy (WMDP) benchmark (Li et al., 2024a) using GPT-4 (Perez et al., 2022) to serve as a proxy measurement for hazardous knowledge in biology, chemistry, and cybersecurity. This process protects against data leakage by never including actual WMDP

    items in the prompt and filters out any questions the base models cannot solve to ensure that the fine-tuned model genuinely has the underlying capability.

2. Using this dataset and framework, we fine-tuned Mistral 7B and LLaMA3 8B with LoRA (Hu et al., 2021). During this fine-tuning process, each WMDP-like evaluation question appears twice: once with the password, where the model is fully elicited and is provided with the correct answer and second where the model is password is absent or locked and the model is provided with an incorrect answer.

3. Using the held-out validation set of generated WMDP-like questions, we recorded the final-token representation from each layer of the fine-tuned model during its forward pass. Specifically, for each prompt $x$, we collect:

$$\text{Act}_\ell^{(\text{locked})}(x) = \text{Hidden}_\ell(x_{\text{locked}})[-1],$$
$$\text{Act}_\ell^{(\text{elicited})}(x) = \text{Hidden}_\ell(x_{\text{elicited}})[-1],$$

We then apply the global layer-wise topological analysis outlined in Section 3.2 to this representation data to study how the model's latent space differs between locked (no password) and elicited (correct password) states.

## F  GLOBAL PH PIPELINE AND RESOURCE CONSTRAINTS

All Vietoris–Rips barcodes are computed with the GPU build of RIPSER++ on a single node equipped with four NVIDIA A100 GPUs (80 GB each). Per layer we draw $K = 128$ independent subsamples of $k = 4096$ activation vectors (64 clean, 64 adversarial). Subsamples are dispatched round-robin to two concurrent RIPSER++ kernels per GPU.

**Memory Footprint.** A complete $k = 4096$ complex truncated at dimension 2 occupies only $2.1 \pm 0.4$ GB of device memory (95th percentile $< 2.8$ GB; Tab. 8), leaving a wide margin inside the 80 GB budget, even when two barcodes are built concurrently on the same GPU.

**Throughput.** The mean walltime per barcode is $36.8 \pm 0.6$s (95th percentile $< 40$s). With four GPUs processing eight barcodes in parallel, a full layer (128 barcodes) finishes in $\approx 10$ min and the five-layer suite of one model in $\approx 50$ min. Running the six models serially therefore completes in about five hours on a single $4 \times$ A100 node—comfortably within the nightly maintenance window.

Table 8: **Computational Costs.** Per-barcode wall-clock time and GPU-memory consumption ($k = 4096$, dimension $\leq 2$). Statistics over $K = 64$ barcodes drawn from the LLAMA-3 8B activations.

| Layer | time $\mu \pm \sigma$ [s] (p95) | memory $\mu \pm \sigma$ [GB] |
|---|---|---|
| 1 | $38.34 \pm 0.76$ (39.6) | $2.27 \pm 0.34$ |
| 8 | $36.79 \pm 0.70$ (38.0) | $2.12 \pm 0.39$ |
| 16 | $36.68 \pm 0.45$ (37.4) | $2.13 \pm 0.30$ |
| 24 | $36.63 \pm 0.71$ (38.1) | $2.03 \pm 0.33$ |
| 32 | $36.62 \pm 0.54$ (37.4) | $2.20 \pm 0.344$ |

After choosing $K = 64$, we recomputed the Monte-Carlo variance $\sigma_f^2$ from the raw, unscaled feature values. For 39 out of 41 statistics, we found $\sigma_f < 0.10$, which would put the standard error SE $= \sigma_f/\sqrt{K}$ below $\Delta^\star/2 = 0.025$ with only $K \leq 20$. The outlier features were those which aggregate counts—total persistence of $H_0$ and the raw count of $H_1$ bars—and need to be transformed for their variance to be directly comparable to the other features. These do not affect the classifier as the features are scaled prior to training and also do not appear as the most informative features for distinguishing between clean and posioned PH-derived features. We conservatively choose $K = 128$ and the resulting ROC–AUCs on the logistic regression model trained only on barcodes are perfect ($1.00 \pm 0.00$), confirming that the subsampling budget is more than sufficient to validate the significance of the features derived from PH, while balancing GPU memory and computation time.

## G  ROBUSTNESS TO ADAPTIVE ATTACKS

We tested the robustness of our identified topological features against real-world attack examples from Microsoft's large-scale `LLMail-Inject` dataset (Abdelnabi et al., 2025) which includes XPIA attack examples and information on their efficacy against four distinct defenses and a fifth setup involving a stacked arrangement of all four. All attack examples are sourced from a public red teaming competition. These attacks are particularly relevant as they include examples that were specifically designed to evade the TASKTRACKER activation-based defense (Abdelnabi et al., 2024), the source of our primary XPIA data. Thus, applying our topological framework to these attack examples is a particularly strong test of whether our topological features represent a fundamental shift in the shape and structure of LLM latent space, or whether it is an artifact that can be easily subverted.

**Methodology.** As the `LLMail-Inject` dataset does not contain paired clean examples, we synthetically generated and manually verified 100 clean counterparts using `Phi-3-medium-4k-instruct`. The small sample size was chosen to ensure that we could verify the quality of the synthetically generated clean examples. We then generated last-token activation data from layer 16 of `Mistral-7B-Instruct-v0.2` for both the clean and adaptive attack inputs and computed the corresponding barcode summary statistics.

**Results.** The topological features of the activation spaces under these adaptive attacks show a clear distinction from the clean examples, as summarized in Table 9. The results show a clear shift towards a simpler, more dispersed topology under adversarial influence.

Table 9: **PH barcode statistics for clean vs. adaptive attack.** Comparison of barcode summary statistics of clean vs. adaptive attack activations from the `LLMail-Inject` dataset on Mistral-7B (Layer 16).

| PH Feature | Clean | Attack (Adaptive) |
|---|---|---|
| H0 Count | 64 | 50 |
| H0 Death Time (Median) | 56.07 | 58.85 |
| H0 Death Time (Mean $\pm$ SD) | $55.43 \pm 21.23$ | $51.73 \pm 28.38$ |
| H1 Count (Loops) | **12** | **4** |
| H1 Birth Time (Median) | **69.36** | **84.92** |
| H1 Death Time (Median) | 71.59 | 86.20 |

- **Fewer, Larger-Scale Loops:** The number of 1-dimensional loops (H1 bars) decreases significantly from 12 in the clean data to just 4 in the adversarial data. Furthermore, their median birth time increases from $\approx 69$ to $\approx 85$, indicating that the remaining topological features are formed at much larger scales.
- **More Dispersed Clusters:** The median H0 death time increases, supporting the hypothesis of greater dispersion. We note that the *mean* H0 death time appears to contradict this trend (decreasing from 55.43 to 51.73). This is due to a small subset of components in the adversarial data merging at very low scales. The median, being more robust to such outliers, better captures the overall geometric shift towards a more spread-out structure.

These findings further suggest that the topological compression signature we identify across models and across XPIA and sandbagging attack conditions reflects a fundamental property of adversarial influence, as the signature remains detectable even against attacks optimized to evade XPIA defenses, including but not limited to the TASKTRACKER activation-based defense.

