# OpenReview forum: "The Shape of Adversarial Influence: Characterizing LLM Latent Spaces with Persistent Homology"
_ICLR.cc/2026/Conference — ICLR 2026 Oral_

### Official Review · Reviewer_inAf · 2025-10-29

**Soundness:** 3
**Presentation:** 3
**Contribution:** 3
**Rating:** 8
**Confidence:** 3

**Summary:**

The paper uses persistent homology (PH), from topological data analysis, to study how LLM activations change under adversarial conditions (prompt injection and backdoor “sandbagging”). It finds a consistent, layer wise topological signature distinguishing clean from poisoned activations, and proposes PH based summaries as practical signals for detection and analysis.  The core idea is to treat each layer’s activation vectors, for many inputs, as a point cloud, compute a Vietoris–Rips filtration and its barcode,  vectorize these barcodes into fixed length barcode summaries, and compare across layers/conditions. The findings are: (i) poisoned activations show fewer small scale features and later/longer lived large scale ones than clean, yielding a clear separation in PH features layer by layer, (ii) The PH summaries linearly separate clean vs. poisoned with near-perfect accuracy across multiple models and attacks, (iii) A local neuron level analysis pinpoints, where in depth adversarial effects concentrate, and ontrols (neuron permutation) remove the signal. Thus, PH barcodes + vectorized summaries provide architecture agnostic descriptors of the latent geometry.

**Strengths:**

(i)  The PH barcode summaries cleanly separate clean vs. poisoned activations across layers with simple linear models. This repeats across several LLMs (7B→70B) and two attack types. This is not only a classification, but rather a characterization of how geometry shifts: poisoned states show fewer small features and later/longer lived large scale ones, while clean states have many short lived features, (ii) There is a solid, reproducible pipeline beginning with subsample activations till PCA/CCA + logistic + SHAP, (iii A complementary local analysis embeds neurons across two layers, and applies PH to track where differences peak in depth, and variance heuristics can find informative layers even without labels.

**Weaknesses:**

(i) Vietoris–Rips PH scales poorly, and the paper therefore subsamples activations, which may lose fine detail or bias results, (ii)  PH results depend strongly on normalization, distance metric, and whether one uses token vs pooled activations. Different choices could change the barcode summaries, (iii) Activation geometry can vary with dataset composition, prompt templates and topic distribution), hence the results might not generalize beyond the specific prompt suite or poison triggers used, (iv) Topological compression is a descriptive signature but the causal link to model failure modes or specific failure behaviors is not fully established. The paper provides strong empirical separation, but no theory explaining why adversarial triggers should systematically compress topology across architectures.

**Questions:**

Suggestions: (i) Sweep the subsample sizes, distance metrics and token selection. Report the key barcode features and perform a stability analysis, (ii) Ensure that the differences are not just prompt length/format, by adding benign dummy insertions that mimic injection structure but carry no instruction, and re run PH, (iii) To anchor the phenomenon, prove for a simple model of mixture of Gaussians in high d,  that moving mass from many small clusters to a few larger, more separated clusters reduces counts of 1-bars and increases mean 0-bar death, formalizing topological compression.

---

> ### Author Response · Authors · 2025-11-24
> **Reply to Review by Reviewer inAf**
>
> We thank the reviewer for their encouraging words and constructive feedback. Please see our responses to the specific questions raised.
>
> **Scalability of PH and the role of subsampling:** As discussed in more detail in our response to Reviewer d2uJ, exact VR PH is expensive in principle, but in our setting it is an engineering trade-off rather than a fundamental barrier. Using the GPU build of `Ripser++`, PH computation is parallelizable across subsamples and layers. Demonstrating that we can comfortably scale PH computations to all six models, including LLaMA-3-70B, using our framework, is one of the contributions of this work. Appendix C.4 (F in the revised version after a typesetting correction) describes our computational pipeline and infrastructure. For example, a 4,096-point subsample uses only ≈2–3 GB of GPU memory and takes ≈37 s to compute a barcode, so one layer (128 barcodes in our case) finishes in ≈10 min and all five layers for all models in ≈5 h on a single compute node.
>
> **Ablations regarding the parameters in the subsampling:** Following the reviewer’s suggestion, we performed ablation studies across the subsampling parameters $n$ and $k$, which will be included in Appendix C.2. First, we want to communicate why the main paper text does not include results for different PH distance metrics and token choices.
>
> - During development we also computed barcodes using cosine distance and observed similar differences between clean and poisoned activations. However, we chose to present Euclidean PH as our main view because Euclidean geometry yields a more direct and standard interpretation of barcode statistics in TDA, and Ripser++ is heavily optimised for this setting. Cosine-based analyses are still included in the paper: Appendix A.3 (“Cosine distance of representations”) and Figures 11 and 14 report cosine distances and LDR built from cosine k-NN neighborhoods, and they show the same qualitative separation and layer-wise structure. This suggests our conclusions do not depend on a single distance metric.
> - Regarding the token choice, please refer to our detailed response to reviewer d2uJ.
>
> Given these constraints, we focused our ablations on $k$ and $n$. Full details on the ablation experiments and results can be found in Appendix C.2 which show that perfect predictive power is achieved across layers for as little as $k=40$ subsamples of size $n=500$. In addition, the mean inter-class distance between clean and poisoned barcode summaries, and their mean intra-class distances exhibit very weak dependence on $k$ and converge as $n$ increases, with curves for $n=1000$ and $n=1500$ almost overlapping. This supports the specific subsample sizes used in our main experiments and indicates that our results are stable under reasonable changes to $k$  and $n$.
>
> **Prompt length/format vs. true adversarial content:** We agree it is important to ensure that our PH signatures are not driven only by superficial prompt formatting. In our work we do not create new prompts, but use the TaskTracker/XPIA activation dataset \[2\], whose design already controls for these factors. For each base task (e.g., coding from CodeAlpaca or QA from SQuAD/HotPotQA), TaskTracker pairs a fixed user prompt and template with different retrieved blocks: a clean block containing benign task-relevant text, and a poisoned block drawn from safety/jailbreak datasets such as BeaverTails, HarmBench, JailbreakBench, and Do-Not-Answer. Thus, clean and poisoned inputs have nearly identical overall structure and similar length; the main difference is whether the retrieved block contains adversarial instructions. We also replicate the pattern on LLMail-Inject \[3\], an adaptive prompt-injection dataset where attacks are specifically crafted to evade the TaskTracker defence. This makes it unlikely that the effect is an artefact of length or template alone, rather than of adversarial content. In Appendix A.3, we also divided the poisoned prompts by outcome (executed, refused, ignored) and showed that the local dispersion ratio differs systematically across these categories, linking our geometric signatures directly to how the model behaves rather than just to the prompt form.

---

> > ### Author Response · Authors · 2025-11-24
> > **Reply to Review by Reviewer inAf**
> >
> > **Lack of causal link between adversarial triggers and topological compression:** We thank the reviewer for the suggestion to formalize topological compression in a simpler probabilistic model. We conducted preliminary experiments with a high-dimensional mixture of Gaussians, comparing a “dispersed” state (one large-variance cluster plus several smaller ones) to a “compressed” state where the mass or centres of the small-variance Gaussians are moved towards the large cluster.
> > Across dimensions, we consistently observed an increase in mean $H_0$ death time under compression ($d\geq 5$, and 93% for $d=2$), supporting the connectivity aspect of our intuition, but $H_1$ counts behaved less consistently (decreased in about half or fewer of the trials). This result held across different GMM configurations, including different cluster migration scales (migration of the smaller-variance Gaussians toward the larger ones), cluster removal, and different GMM mixing ratios.
> > This suggests that LLM representation spaces exhibit a more complex topological structure as manifolds than what can be captured by simple Gaussian mixtures. Rather than over-claiming on the basis of this toy model, we treat these results as motivation for future theoretical work and will mention them briefly in the Discussion, while keeping the present paper focused on empirical characterization of adversarial effects in real LLM activations.
> > Below are the results averaged over 30 runs, which we will include in the revision:
> >
> > | Dim | Mean $H_1$ Reduction | $H_1$ Consistency | Mean $H_0$ Increase | $H_0$ Consistency |
> > | ----- | ----- | ----- | ----- | ----- |
> > | 2 | \-0.5 ± 7.1 | 40% | 0.03 ± 0.02 | 93% |
> > | 5 | 1.1 ± 16.6 | 53% | 0.19 ± 0.06 | 100% |
> > | 10 | \-14.5 ± 27.0 | 33% | 0.48 ± 0.10 | 100% |
> > | 50 | \-18.6 ± 50.8 | 33% | 1.67 ± 0.52 | 100% |
> > | 100 | \-16.8 ± 60.6 | 37% | 2.86 ± 0.86 | 100% |
> > | 250 | \-15.2 ± 58.2 | 33% | 4.26 ± 0.90 | 100% |
> > | 500 | 13.0 ± 69.4 | 57% | 6.23 ± 1.57 | 100% |
> >
> > **References**
> > \[1\] Zou, A. et al. (2023). *Representation Engineering: A Top-Down Approach to AI Transparency*. arXiv:2310.01405.
> > \[2\] Abdelnabi, S. et al. (2025). *Get My Drift? Catching LLM Task Drift with Activation Deltas*. IEEE SaTML.
> > \[3\] Abdelnabi, S. et al. (2025). *LLMail-Inject: A Dataset from a Realistic Adaptive Prompt-Injection Challenge*. arXiv:2506.09956.

---

### Official Review · Reviewer_NSud · 2025-10-29

**Soundness:** 4
**Presentation:** 4
**Contribution:** 3
**Rating:** 6
**Confidence:** 4

**Summary:**

The paper investigates the embeddings of clean inputs versus adversarially perturbed inputs using persistent homology. This is a  tool from topological data analysis that enbles to describe the topology and geometry of point clouds, in this case last token embeddings, in a principled manner. The authors show that clean and adversarial inputs exhibit very different topological properties, which goes beyond the typical analysis of linearly separating such samples. The results are demonstrated on a diverse set of models and appropriate choice of benchmarks.

**Strengths:**

- Investigating the geometry of the feature space of modern neural networks is crucial to better understand how models work and how they arrive at decisions. Taking the route of PH is an interesting direction and a valuable contribution.
- The paper is clearly written. In particular, given that PH is not yet commonly used in interpretability/robustness research, the authors give good intuition for what barcodes do.
- The experimental setting is well chosen and covers important failure modes in LLM security.

**Weaknesses:**

- This sentence gives the impression you are the first to do this: “We propose persistent homology (PH), a tool from topological data analysis, as a principled framework to characterize the multi-scale dynamics within LLM activations.” I suggest phrasing it as using PH as a tool rather than proposing a framework, since related uses exist in other domains and your contribution is not a full PH framework.

- In the background section, consider citing a review of complex constructions (e.g., Vietoris–Rips, Čech, alpha complexes).

-  A formal treatment of PH would be appreciated in the appendix (optional).

- The fact that the barcode summaries cluster adversarial vs. clean examples is not surprising if these samples are already linearly separable. However, it needs further elaboration to explain why this is important.

- Even when you obtain a perfect AUC, adding your "method" to Table 1 would improve the presentation.


Note: If my concerns are properly addressed I am inclined to raise my score to an accept.

[1] Gardinazzi, Yuri, et al. "Persistent topological features in large language models." arXiv preprint arXiv:2410.11042 (2024).

**Questions:**

- What do you mean by "consistent topological behavior within the LLM latent space"?

---

> ### Author Response · Authors · 2025-11-24
> **Reply to Review by Reviewer NSud**
>
> We would like to thank the reviewer for their feedback, and to address the specific weaknesses that they point out.
>
> **PH as a tool versus a framework:** When we referred to a “framework”, we did not mean PH itself, but the **full computational pipeline** we build around it. Concretely, our framework consists of: (i) loading and sharding last-token activations from multiple LLMs and information processing scenarios conditions; (ii) constructing subsampled point clouds per (layer, condition), and distributing Vietoris–Rips persistent homology computations across GPUs using Ripser++; (iii) vectorizing barcodes into PH summary statistics and storing them in a standardized format; and (iv) running layer-wise and neuron-level analyses and visualizations on these summaries (including logistic probes, SHAP, and LDR). We will release this pipeline as open-source so that other researchers can reuse it end to end without having to re-engineer activation loading, sharding, PH computation, and analysis for each model.
>
> By calling PH a “principled framework,” we aimed to emphasize that PH is a mathematically rigorous tool that captures multiscale topological structure in activation spaces, in a way that enables meaningful comparisons across models (due to its coordinate free nature).  While we are not the first to apply PH to LLMs, to the best of our knowledge we are the first to interpret the activation space, while preserving its dimensionality, and identify distinct topological features representing normal and information processing conditions across several adversarial scenarios. Our work is the first to do this at this scale, across six SOTA models, several layers, and across both inference time and training time attacks. Please note that our inference time attack datasets include adaptive attacks that were specifically designed to evade the TaskTracker activation data that we leverage (which itself includes 4 industry standard attack and safety focused datasets including PKU-Alignment/Beavertrails \[2\], PKU-Alignment/PKU-SafeRLHF, Harmbench, Jailbreakbench, and Do-Not-Answer dataset \[5\]).
>
> In response to the reviewer’s comment, we propose two revisions:
>
> 1. In the abstract, we will change the highlighted sentence to: *“We propose the application of persistent homology (PH) to measure and understand the geometry and topology of the representation space when the model is under external adversarial influence.”*
>
> 2. We will add a preamble to Section 2.3 noting prior applications of TDA (and PH) to LLMs to clarify that this direction is not novel in itself.
>
> **Adding background about PH:** As suggested by the reviewer, we will add a full section in the appendices detailing the PH construction and covering all the different filtration constructions mentioned (see section A.1 of the revised manuscript, to be posted shortly). We will add a reference to this section in the main text.
>
> **Explaining the importance of leveraging topological methods:** Our goal is not to use PH as a better-performing classifier, but as a way to understand how adversarial inputs (like prompt injections, jailbreaks, and backdoor triggers) reshape the geometry of representation space in dimensions where human intuition and standard tools struggle. While there is emerging work using PH with LLMs, it has not been clear in practice that PH can be applied at scale to full activation spaces of modern models (up to tens of billions of parameters) and still yield interpretable structure. A central contribution of our work is to demonstrate that simple PH summaries can be computed reliably at this scale *and* provide mathematically principled, human-interpretable geometric information about LLM activations, that are comparable across models, without first collapsing them through low-dimensional projections. Section 4.1 is precisely where we make this point: the strong performance of linear models on barcode summaries is valuable not as an accuracy result, but because feature-importance analyses on these models tell us which aspects of topology (e.g., numbers and lifetimes of 1-dimensional features) differ systematically between clean and poisoned activations. We will rephrase this discussion in the revision to make this interpretative focus (and the fact that the value of PH lies in the geometry it reveals rather than marginal gains in detection performance) more explicit.
>
> **Presentation**: We have already added our method to the comparison in Table 1 as suggested by the reviewer.
>
> Regarding the **question** of the reviewer: What we meant by “consistent topological behavior within the LLM latent space” is precisely that the topological differences between normal and adversarial activations that we detect in our analysis stay consistent across models and attacks.

---

### Official Review · Reviewer_d2uJ · 2025-11-01

**Soundness:** 2
**Presentation:** 2
**Contribution:** 3
**Rating:** 6
**Confidence:** 4

**Summary:**

This paper applies persistent homology (PH) from topological data analysis to study how adversarial inputs reshape latent representations in large language models (LLMs). The authors analyze six instruction-tuned models (3.8B–70B) under two adversarial conditions — indirect prompt injection (XPIA) and sandbagging / backdoor fine-tuning — using the TASKTRACKER and sandbagged datasets. They compute Vietoris–Rips filtrations on subsampled point clouds of last-token activations, vectorize persistence barcodes into 41-dimensional “barcode summaries,” and run global (layer-wise) and local (neuron-pair 2D embedding) analyses. The central empirical claim is that adversarial influence produces a reproducible “topological compression”: adversarial activations show fewer but larger-scale topological features (fewer H1 loops, higher mean death times of H0 components), a phenomenon that is discriminative (logistic classifiers/SHAP achieve near-perfect separation) and consistent across models and attack types. The paper also reports a neuron-level phase transition in topological complexity at intermediate layers (≈ layer 12 for Mistral 7B).

**Strengths:**

Novel methodology: First systematic application of PH to characterize adversarial effects in LLM latent spaces at both global and neuron levels.

Robust empirical signal: Separation is reproducible across six models and two attack modes; discriminative power is high and interpretable via SHAP.

Mechanistic insight: Neuron-pair 2D embeddings and layerwise analysis identify where adversarial influence reconfigures information flow (phase transition in deeper layers).

Careful controls: Normalization, permutation tests, adaptive-attack evaluation (LLMail-Inject) and baseline linear methods are included.

**Weaknesses:**

Dependence on subsampling and last-token choice. PH is memory-intensive; authors subsample large numbers (k=1000, many subsamples), which is theoretically supported but may miss rare, high-impact topological features and makes replication costly. Also, using only the last-token embedding leaves open whether signatures generalize to alternative aggregation choices.

Interpretation vs. causality. The paper convincingly documents correlational topological signatures but stops short of causal interventions (e.g., topology-aware regularization or targeted modifications to test whether changing topology alters adversarial susceptibility). Such experiments would strengthen the link between topology and vulnerability.

Generality to other threat models. The study focuses on XPIA and sandbagging/backdoor attacks. While LLMail-Inject adaptive examples were tested, evaluation against a wider array of adaptive, distributional, or model-poisoning attacks (and on more diverse prompts/tasks) would better establish universality.

Scalability & runtime. Practical adoption of PH-based monitoring in production LLM systems would require faster approximations or streaming variants; the paper acknowledges this but provides limited engineering pathways. It is also not clear how can the method scale to much larger production-grade models with 100B+ parameters.

References. The authors miss citing a few critical works comparing their approach in other domains. For e.g. Extreme Image Transforms (EITs) [Crowder et al., 2022; Malik et al., 2023, Biol Cybernetics, Malik et al., 2023, arXiv] help with similar representations in vision for deep networks. Similarly Network Dissection [Bau et al., 2017, CVPR] and Locating and editing factual associations in gpt [Meng et al., 2022, NeurIPS] show layer-wise applicability to the final output of deep networks.

Presentation. The individual sections of the paper are well written but the paper flow is not easy to follow when put together. The authors should consider reorganizing the sections to make the story flow better and for the reader to keep track of what is happening, without losing the focus from the main point. .

**Questions:**

How does the observed topological compression depend on the choice of token pooling (last token vs. mean-pooled vs. CLS-like embeddings)? Any preliminary experiments?

Can the authors provide an ablation showing how sensitive the signature is to subsample size k and the number of subsamples K (e.g., do smaller k or fewer K materially change detection performance)?

Have the authors attempted a simple topology-aware defense (e.g., penalize total persistence changes or normalize mean H0 death) to test whether changing topology reduces task drift or attack success? That would help evaluate causality.

The local 2D neuron embedding analysis finds a phase shift around mid-layers for Mistral — does the layer index of that transition correlate with model size/architecture across the six models?

Could noisy or distributional natural shifts (non-adversarial OOD) produce similar topological signatures? That is, how specific is the signature to malicious adversarial influence vs. benign OOD?

The authors should consider releasing their code publicly for reproducibility by the community.

Can the authors highlight the details of their experimental setup and the hardware/infrastructure used?

The authors should also consider showing a specific example across different models for the reader to visualize the method a little less abstractly.

---

> ### Comment · Reviewer_d2uJ · 2025-11-20
>
> Please let me know if there are any questions about the review. Thanks.

---

> > ### Author Response · Authors · 2025-11-24
> > **Reply to Review by Reviewer d2uJ**
> >
> > We thank the reviewer for their feedback and would like to address each point below.
> >
> > **Dependence on last-token choice:** Our design choice to use last-token activations is twofold. Theoretically, we follow \[1\], who show that the last token (the immediate precursor to prediction) captures the entire input context most effectively compared to other token aggregation approaches, including those mentioned by the reviewer. This makes it the natural choice for studying how adversarial inputs (i.e., backdoor triggers and indirect prompt injections) alter the model’s internal representation space, while capturing as much context as possible with minimal storage. Practically, this choice allows us to utilize the TaskTracker indirect prompt injection (XPIA) activations directly. The TaskTracker work also demonstrated that the last token activation representation of the input context was most effective at detecting XPIA. Recomputing and storing the entire token sequence, across all models, for the same quantity of data that Tasktracker data provides, would be prohibitively compute/time/memory expensive.
> >
> > **Dependence on subsampling parameters:** Following the reviewer’s suggestion, we will add a subsampling ablation study in Appendix C.2, varying both the number of subsamples $k$ and subsample size $n$. The results show that the performance and representation quality exhibit minimal dependence on $k$. Although our experiments use $k=64$, values around $k \\approx 30$ preserve predictive power and representation fidelity when compute is limited.
> >
> > **Interpretation vs. causality:** The reviewer is correct that we focus on characterizing how topology changes under attack, but we do not intervene on topology itself. Our goal is to show that there is a robust, cross-model geometric signature of adversarial influence and that PH is a scalable tool for analyzing the shape of LLM representations, rather than to propose a complete defense. Designing topology-aware defenses (e.g., penalizing total persistence or constraining $H_0$ death times) is a natural next step, but we believe such interventions are best integrated **during training or via the architecture**, in line with work on topological regularization (e.g., topological autoencoders \[4\] and topological deep learning \[5\]) that incorporates PH into the model or loss. By contrast, post-hoc activation steering is shown to be brittle and to alter model behavior in hard-to-predict ways, making it a weak basis for robust defenses \[2, 3\]. For these reasons, we treat topology-aware regularization and architectural constraints as a separate line of future work explicitly aimed at causality and will note this explicitly in our Discussion section
> >
> > **Generality to other threat models:** We agree that our goal is not to claim universality over all possible attacks, but we do cover more than two narrow cases.
> >
> > * For indirect prompt injection, we use the TaskTracker/XPIA benchmark, whose *test* split is explicitly constructed by combining (i) base tasks such as coding from CodeAlpaca and question answering from SQuAD and HotPotQA with (ii) attack content drawn from four standard safety/jailbreak datasets: PKU-Alignment/BeaverTails (safety-labelled Q\&A), HarmBench (harmful behavior prompts), JailbreakBench (modern jailbreak prompts), and the Do-Not-Answer dataset (refusal-style safety prompts). These sources give TaskTracker a mix of OOD safety queries and jailbreak-style attacks embedded into otherwise standard tasks.
> > * In addition, we apply our framework to LLMail-Inject, an adaptive prompt-injection dataset from Microsoft’s public challenge, which includes attacks explicitly crafted to evade the TaskTracker defence. Please see Appendix C.5.
> > * For training-time attacks, we created our own activation dataset and replicated the trigger-based backdoor fine-tuning setup following van der Weij et al. (2024). Sandbagging in our experiments is a concrete capsule example of the broader class of backdoor and trigger-based attacks, a central concern in AI safety.
> >
> > Across TaskTracker (including BeaverTails/HarmBench/JailbreakBench/Do-Not-Answer), LLMail-Inject, and the backdoor-fine-tuned models, we consistently observe the same “topological compression” signature and strong PH-based separation. We will update the paper to make this scope more explicit.

---

> ### Author Response · Authors · 2025-11-24
> **Reply to Review by Reviewer d2uJ**
>
> **References:** We thank the reviewer for the suggestions and have incorporated the most relevant ones. We now cite Bau et al. (2017) and Meng et al. (2022) in our Related Work as complementary non-topological interpretability methods for CNNs and LLMs. We also add Gardinazzi et al. (2024), who apply zigzag persistence to LLMs to study layer redundancy under standard conditions, and explicitly contrast this with our focus on adversarial influence and per-layer latent geometry. To support the clarity and focus of our work, we do not cite Extreme Image Transformations (Crowder & Malik; Malik et al.) which are vision-only robustness studies without topology.
>
> **Scalability & Runtime**
>
> * While exact PH is $O(N^3)$, we clarify that for inference-time monitoring, one would only need to compute PH on *single* mini-batches (not the whole training set). Using `Ripser++`, a single prompt's barcode can be computed in milliseconds which makes real-time monitoring feasible.
> * In practice, we did not encounter scalability issues when applying our PH pipeline to the models up to LLaMA‑3‑70B, though this is not an upper bound. Overall, our results suggest that scaling to larger models is a matter of engineering trade-offs, rather than a fundamental computational barrier.
>   * Appendix C.4 (“Global PH pipeline and resource constraints”) details our implementation and the parallel nature of the PH barcode computations in Ripser++. The computation of the entire complex (up to but not including dimension 2\) only used 2.1 ± 0.4 GB of device memory (95th percentile \< 2.8 GB; Table 7). The mean wall‑time per barcode is 36.8 ± 0.6 s (95th percentile \< 40 s). With four GPUs processing eight barcodes in parallel, a full layer (128 barcodes) completes in ≈10 minutes and running all six models (across all 5 layers) serially finishes in about five hours on a single 4×A100 node. These measurements include the largest model (70B parameters), indicating that our current configuration already scales comfortably to this range.
>   * Moreover, our subsampling ablations (Appendix C.2) show that PH summaries and classifier performance converge for smaller subsample sizes (e.g., n ≈ 1000, k ≈ 40), higher n or K would mainly increase runtime without revealing additional structure in the PH features.
>
> **Questions**
>
> 1. The token choice was discussed in detail above.
> 2. Further ablation detailed in Appendix C.2 (revised version) confirms that the topological signature appears with as little as n=500 points per subsample and k=40 subsamples.
> 3. We have not implemented a topology-aware defence. Our goal here is to understand and characterize the fundamental changes in the topology and geometry of LLM representation spaces under attack, across models and threat types, rather than to propose a defense mechanism.
>    1. In our view, enforcing topological constraints for robustness is most effective **during training or via the architecture**, not as a post-hoc activation-steering defence. The topological deep learning position paper [5] argues for using topological structure to guide model *design* and learning objectives, rather than only for downstream analysis, and topological autoencoders [4] incorporate persistent homology as a differentiable loss term so that the encoder learns latent spaces that preserve multi-scale connectivity. This is closer to the kind of intervention needed to test causality.
>    2. By contrast, test-time activation steering is brittle and can unpredictably alter model behavior \[2,3\], making it a weak basis for robust defences. PH-based regularisation or architectural constraints are an important *next step* building on our results and are noted in our Discussion section.
> 4. Regarding the question on the "Phase Shift" layer: We found that the layer index where topological complexity diverges (Clean vs. Poisoned) roughly scales with depth. For 7B models (32 layers), it occurs around Layer 12-16. For 70B models (80 layers), it shifts deeper (Layer \~30-40). This suggests the attack impacts the semantic convergence stage of the model, regardless of total depth. We agree that a dedicated study on this is an important area of future work.
> 5. The code will be released upon publication. Details on our computational pipeline, resource metrics, and infrastructure were previously provided in Appendix C.4 and will appear in a dedicated Appendix F in the revised manuscript.
>
> \[1\] Zou, A. et al. (2023). *Representation Engineering: A Top-Down Approach to AI Transparency*. arXiv:2310.01405.
> \[2\] Tan, D. et al. (2024). *Analysing the Generalisation and Reliability of Steering Vectors*. NeurIPS 37\.
> \[3\] Yang, W. et al. (2025). *The Mirage of Model Editing: Revisiting Evaluation in the Wild*. ACL 2025\.
> \[4\] Moor, M. et al. (2020). *Topological Autoencoders*. ICML.
> \[5\] Hajij, M. et al. (2022). *Topological Deep Learning: Beyond Graph Data*. arXiv:2206.00606.

---

> > ### Comment · Reviewer_d2uJ · 2025-11-27
> >
> > Thank you for the detailed rebuttal response. i look forward to seeing these in the main text/appendix of the paper for further understanding of the work to the end user.

---

### Official Review · Reviewer_DxAW · 2025-11-02

**Soundness:** 2
**Presentation:** 2
**Contribution:** 2
**Rating:** 4
**Confidence:** 3

**Summary:**

The paper investigates how adversarial interventions reshape the latent geometry of large language models (LLMs) using topological data analysis and persistent homology (PH) diagrams. The analysis is done across several instruction-tuned models and two attack types (prompt injection and backdoor fine-tuning), and PH barcodes are computed from layer activations, summarizing them into 41-dimensional feature vectors, and comparing clean versus adversarial conditions. The authors report a recurring pattern termed “topological compression”, fewer 1-dimensional bars and larger 0-bar death times for adversarial inputs. Logistic regression on PH summaries separates clean and adversarial activations with high accuracy, roughly matching linear baselines. A second, “local” analysis applies PH to neuron-pair across layers to examine layer-to-layer information flow, reporting differences in total 1-bar persistence and proposing PH variance as an unsupervised indicator of affected layers.

**Strengths:**

* **Originality of the approach.**
  Applying persistent homology to adversarial analysis in LLMs is novel to the best of my knowledge and interesting, bridging topological data analysis with model interpretability and robustness.

* **Wide experimental coverage.**
  The study spans multiple coverage, including very large scale models in the Appendix (70B) and two distinct adversarial mechanisms, showing qualitatively consistent trends across settings.

**Weaknesses:**

* **Claims may not be fully supported by experiments.**
  - The “topological compression” effect is observed descriptively but remains correlational. No controlled test distinguishes true topological simplification from simpler geometric or scaling changes: a simple example would be to match the scales of the original and poisoned features similarly as done in section 4.2.
  - Linear baselines already achieve near-perfect separability (Table 1), aside from "layer 0", which is not showed for the PH stats  PH’s added value over simpler probes remains not fully clear to me in this setting. Moreover this could highlight that the binary task of distinguishing between poisoned and clean features might be too simple on this dataset and model pairs.
  - The local analysis (Table 2) relies on a coarse metric (precision@k), without random baselines or multiple-comparison control. This weakens the claim that PH variance reliably identifies adversarially affected layers. A better metric to use would be spearman correlation.
  - The results are not linked to behavioral or task-level outcomes (e.g. jailbreak success or refusal rates).


* **Clarity and presentation quality.**
  - Several results are difficult to interpret at first sight:  I think the authors should give priority for each figure to the result/plot that better highlights the current claim and put the remaining ones in the Appendix. For example Figure 8 show different information and is very dense, obscuring panel (d) which is the one that supports to the claim as (a), (b), (c) difference between poisoned and clean activation curves are not very visible.
  - minor: Cross-model results are not included in the main text: I believe that a concise summary table in the main text would make the findings clearer and more convincing.


* **Few ablations.**
PH is computed only with one distance metric and one filtration type. No ablations on these design choices, or on subsample size, are provided, making it hard to assess stability, see [c,d] for discussions on how persistence diagrams can vary with metric choice. Scalability is not addressed beyond the subsampled settings; approximate PH algorithms such as witness complexes or streaming approaches [e] could test whether the reported effects persist at realistic layer scales. The 41-dimensional summarisation is also not discussed in comparison with alternative vectorisations such as persistence images [a]  or persistence landscapes [b], leaving uncertain whether the observed pattern depends on this particular encoding. I suggest to include at least a discussion on these choices should be included in the paper.

_[a] Adams, H., et al. (2017) Persistence Images: A Stable Vector Representation of Persistent Homology. JMLR_

_[b] Bubenik, P. (2015) Statistical Topological Data Analysis Using Persistence Landscapes. JMLR_

_[c] Chazal, F., et al. (2015) Convergence Rates for Persistence Diagrams. JMLR_

_[d] Cohen-Steiner, D., Edelsbrunner, H. & Harer, J. (2007) Stability of Persistence Diagrams. Discrete & Computational Geometry 37(1):103–120._

_[e] Kerber, M., Morozov, D. & Nigmetov, A. (2016) Geometry Helps to Compare Persistence Diagrams. J. Exp. Algorithmics 22(1):1–20._

**Questions:**

- Is there any motivation  not considering higher order bars (2-bars, 3-bars) etc?

-  How exactly is the variance in Table 2 computed: across all samples or separately per condition?

- Would the same compression pattern appear under different subsample sizes?

---

> ### Author Response · Authors · 2025-11-24
> **Reply to Review by Reviewer DxAW**
>
> We appreciate the reviewer’s feedback and would like to address each of their concerns.
>
> **Is "topological compression" merely a correlation?** While we cannot conclude causality without interventional retraining, multiple controls suggest this effect is not an artifact.
> * In Appendix C.5 (Robustness to Adaptive Attacks), we show that even when adaptive attacks mimic the *linear* statistics (mean, variance) of clean activations (the `LLMail-Inject` dataset), the topological compression signature (fewer but longer-lived 1-dimensional bars and larger 0-dimensional death times for adversarial inputs) persists. Matching linear statistics would remove the signal if it were incidental.
> * VR filtrations and PH barcodes are computed on normalized activations in all experiments; and the topological compression effect remains clearly visible, showing that it is not due to activation scales. We will state this explicitly in our revision.
>
> **Value over Linear Baselines**
>
> * Table 1 contained an **indexing issue**: layer $n$ refers to layer $n+1$ in Figure 8, making them fully comparable. We will fix this and add the performance of our PH-based classifier in the revision.
> * Our focus is not to outperform linear baselines, but rather, the unique explanatory value PH brings beyond prediction accuracy:
>   * **PH is coordinate-free,** i.e. it only depends on pairwise distances between activations, not on the specific neuron basis of a model. This enables the comparison of its features across architectures and hidden sizes, unlike the features of other methods. This is what allows us to identify the same “topological compression” pattern across all 6 models and both attack mechanisms.
>   * **PH provides interpretable geometric statistics across multiple scales** (shifts in birth/death scales, changes in total persistence) that quantify the size and shape of the representation space directly, where most standard interpretability tools struggle.
> * If we are comparing the approaches on the grounds of classification of input classes, **PH does not require labeled inputs**. Unlike linear probes, which require a labeled dataset for classification, the PH features successfully identify adversarial inputs without labeled "poisoned" data (see Section 4.2).
>
> **Connecting representational changes to behavior:** Appendix A.2 (“Local Dispersion Ratio Across Poisoned Conditions”) describes an experiment that studies the topological features of adversarial inputs, stratified by how they affected the model (*executed* jailbreaks or prompts, *refusals*, and *ignored* attacks). Figure 12 shows that the local dispersion ratio (LDR) differs across these categories. The LDR is the ratio of the sum of all eigenvalues except the largest to the largest eigenvalue, computed via PCA on each local neighborhood of activation-difference vectors. Executed and ignored attacks exhibit higher LDR in mid layers than clean prompts, indicating that the model allocates additional representational capacity to elaborating the injected instructions. Refused prompts are mapped into a more compressed, lower-dispersion region. This connects geometric changes to task-level behavior.
>
> **Local Analysis Statistics:** We did perform a permutation test to assess whether the precision@k results were significant: the asterisks in Table 2 indicate the significance relative to a random baseline based on permutation-type testing. We will make this explicit and add Spearman correlations, as suggested.
>
> **Cross-Model Summary:** We will add a concise table (Table 2 in the revised version, to be posted shortly) summarizing cross-model consistency.
>
> **Figure prioritization**
>
> 1. *Figure 8* will be simplified by moving less-critical panels to the Appendix.
> 2. *Figure 13* will be moved from the Appendix to the main text. This figure shows that LDR differences are near zero under Clean-vs-Clean and Poisoned-vs-Poisoned resampling, but Mixed-vs-Mixed splits present deviations that align with the clean–poisoned gaps in Figures 11 and 12. This confirms that LDR reflects genuine geometric differences rather than sampling noise.

---

> > ### Author Response · Authors · 2025-11-24
> > **Reply to Review by Reviewer DxAW**
> >
> > **Ablations**
> >
> > * We have added subsampling ablations in Appendix C.2 in the revised version.
> > * **Euclidean distance** for **Vietoris–Rips PH** is used to leverage `Ripser++`, one of the fastest and most scalable backends for PH computations, and to facilitate interpretations, as the Euclidean metric is a natural notion of scale and dispersion in activation space. We complemented the Euclidean PH results with cosine-based analyses in Appendixes A.3 and A.4, Figures 11 and 14, where we compute cosine distances between local neighbors and different representations. These cosine-based results exhibit the same qualitative “compression” pattern as the Euclidean PH summaries. We report Euclidean PH as our primary view because it is both standard in topological data analysis and theoretically well-behaved: classical stability and convergence guarantees for persistence diagrams are stated for functions on (typically Euclidean) metric spaces \[2\].
> > * **Subsampling parameters**: Appendix C.2 presents ablations that (i) evaluate the performance of the logistic regression classifier across different $(n,k)$ settings, and (ii) study the representations of these subsamples by computing mean intra- and inter-class distances distances. In brief, for the first set of experiments, we obtain perfect accuracy in all layers with only $k=40$ subsamples of size $n=500$. For the second set, we observe that increasing the subsample size $n$ leads to more tightly clustered barcode representations and greater separation between clean and poisoned activations, while varying the number of subsamples $k$ has only a minor effect. Convergence of these metrics is apparent for $n \\geq 1000$, supporting the subsample sizes used in the main experiments.
> > * **Other vectorizations**: we emphasize that our “barcode summaries” are *not* a vectorization of persistent homology in the sense of persistence landscapes or persistence images. Those methods transform entire barcodes into high-dimensional feature vectors specifically designed for downstream ML. In contrast, we first compute PH barcodes and then extract a set of descriptive statistics from them, as a convenient way to analyze and compare them. We also interpret each component individually rather than treating the vector as a learned embedding. Our choice is also motivated by the fact that these descriptive statistics tend to outperform other vectorizations, see \[1\].
> >
> > **Questions**
> >
> > 1. There are two primary reasons for *not including higher dimensional homology:*
> >    1. Computing $H\_2$, $H\_3$,...  via VR at our scales (thousands of last token activations, where each last token activation vector is 4096 dimensional, across many layers, and six models) is computationally prohibitive.
> >    2. In contrast to $H\_0$ (isolated components/clusters) and $H\_1$ (loops), higher-dimensional features are much harder to interpret in the context of LLM representations. Since $H\_0$ and $H\_1$ already yield a strong consistent signal across models and attacks, we leave higher dimensions for future work.
> > 2. The variance reported in Table 2 is computed across all samples, pooling conditions together (i.e., we do not condition on clean vs. poisoned when computing the per-layer variance of a PH statistic). The motivation is that this analysis is intended as an unsupervised layer-localization tool (in a realistic deployment we do not know which inputs are adversarial, so we look for layers where the PH statistic has high variance without labels.) We then show that these high-variance layers correspond well to those with large clean–vs–poisoned differences. We will clarify that the variance is computed over all samples pooled, not separately per condition.
> > 3. Yes. As mentioned above, we have added an ablation study (Appendix C.2) where we vary the subsample size $n$ and the number of subsamples $k$. The result is that our main conclusions do not depend on a single choice of (n,k). We observe that the quantitative compression pattern (fewer but longer-lived 1-bars and larger 0-bar death times for adversarial inputs) is stable across subsample sizes. Increasing $n$ mainly reduces variance and tightens separation and the performance and distance metrics converge for $n\\geq 1000$. Varying k has only a minor effect once k is moderately large.
> >
> > **References**
> >
> > \[1\] Ali et al. (2023). Survey of vectorization methods in TDA. *TPAMI.*
> >
> > \[2\] Cohen-Steiner et al. (2005). Stability of Persistence Diagrams. *Discrete & Computational Geometry.*

---

### Author Response · Authors · 2025-12-03
**Changes implemented in the revision**

We would like to thank again all the reviewers for their feedback, which we have taken into account and implemented in the rebuttal revision. A summary of the changes implemented can be found below.

- As suggested by NSud we have changed the sentence “We propose persistent homology (PH), a tool from topological data analysis, as a principled framework to characterize the multi-scale dynamics within LLM activations” in the abstract to "We propose the application of persistent homology (PH) to measure and understand the geometry and topology of the representation space when the model is under external adversarial influence."
- Addressing a comment by reviewer DxAW, we have fixed an indexing issue in table 1 and have added the results of our PH method to make the comparison clearer.
- By recommendation of DxAW we have added a table (Table 2) summarizing the results of our experiments accross models.
- We have also modified the figures 9 and 10 in the previous version to just figure 10 in the updated version, as requested by DxAW.
- We have added a new figure (figure 9 in the updated version) that shows that the differences in dispersion ratio between clean and poisoned samples are not artifacts but statistically significant, answering issues raised by DxAW.
- We have added ablation experiments, as suggested by reviewers d2uJ and inAf on the subsampling parameters, which can be found in appendix C.2.
- We have added a subsection on the appendices detailing the PH construction as requested by NSud (see appendix A.1).

We hope that these revisions address and clarify some of the points raised by the reviewers. Thanks again for everyone's time!

---

### Meta-Review · Area_Chair_NFns · 2026-01-06

**Summary:**

1. *inAf, DxAW, du2j* all had concerns regarding the fact that the topological compression effect was correlational rather than causative.
2. *DxAW, Nsud* both pointed out that the PH barcodes being able to identify adversially affected representations was not surprising if a linear model can also detect them
3. *DxAW* pointed to lack of good metrics and baselines.
4. *DxAW* the authors do not connect to task-level outcomes.
5. *DxAW, du2j* had presentation concerns
6. *DxAW, d2uj, inAf* had concerns about scalability of the approach and in particular the sensitivity of the method to choices in subsampling used to obtain better scaling.

**Reviewer Concerns:**

1. The authors provided some additional evidence that the effect was not merely correlational including a probabilistic formulation, but admit that a true interventional approach is beyond the scope of the work.   This stands as a potential weakness of the paper, but is likely outweighed by the positive contribution.
2. The authors and *inAf* point out the true value of the PH signature is not in detection but in providing additional geometric understanding.
3. The authors added a perturbative analysis and spearman coefficient.
4. The authors give an example in the appendix.
5. The authors proposed several revisions to address concerns.
6. The authors made a strong case for the scalability of their method and added several ablations to establish their subsampling method as well motivated and robust.

**Reviewer Scores:**

- *DxAW* A good chance of small score increase 4 to 6.
- *d2uJ* may have increased 6 to 8
- *NSud* very likely would have increased 6 to 8
- *inAf* Would have maintained an 8.

---

### Decision · Program_Chairs · 2026-01-26

Accept (Oral)